# Neural Collapse To Multiple Centers For Imbalanced Data

Hongren Yan, Yuhua Qian*, Furong Peng, Jiachen Luo, Zheqing Zhu, Feijiang Li

Shanxi University

## Abstract

Neural Collapse (NC) was a recently discovered phenomenon that the output features and the classifier weights of the neural network converge to optimal geometric structures at the Terminal Phase of Training (TPT) under various losses. However, the relationship between these optimal structures at TPT and the classification performance remains elusive, especially in imbalanced learning. Even though it is noticed that fixing the classifier to an optimal structure can mitigate the minority collapse problem, the performance is still not comparable to the classical imbalanced learning methods with a learnable classifier. In this work, we find that the optimal structure can be designed to represent a better classification rule, and thus achieve better performance. In particular, we justify that, to achieve better classification, the features from the minor classes should align with more directions. This justification then yields a decision rule called the Generalized Classification Rule (GCR) and we also term these directions as the centers of the classes. Then we study the NC under an MSE-type loss via the Unconstrained Features Model (UFM) framework where (1) the features from a class tend to collapse to the mean of the corresponding centers of that class (named Neural Collapse to Multiple Centers (NCMC)) at the global optimum, and (2) the original classifier approximates a surrogate to GCR when NCMC occurs. Based on the analysis, we develop a strategy for determining the number of centers and propose a Cosine Loss function for the fixed classifier that induces NCMC. Our experiments have shown that the Cosine Loss can induce NCMC and has performance on long-tail classification comparable to the classical imbalanced learning methods.

## 1 Introduction

Deep neural networks are popular choices classification tasks[1, 2, 3, 4, 5, 6, 7]. Researchers try to demystify the deep representations learned from data [8, 9]. A recent paper [10] observed "Neural Collapse" (NC) phenomenon: all the backbone network output features from each class converge into their corresponding vertices of an equiangular tight frame (ETF) and the within-class variability collapses.

The layer-peeled model (LPM) [11] and unconstrained feature model (UFM) [12] are the simplified model to study NC, in which the backbone output feature and the classifier weights are assumed free variables to optimize. NC phenomena occur with different loss functions. The optimality of UFM satisfies NC under the CE loss with constraints [13, 11, 14], regularization [15], or even no explicit constraint [16]. MSE objectives also induce NC at global optimality [12, 17, 18, 19]. There is another line of works that extend UFM or LPM on deeper linear layers [19, 20, 21, 22].

Data imbalance has been recently considered in NC literature[11, 23, 20]. In particular, Fang et al. [11] originally observe the "minority collapse" phenomenon that the classifier weights of the minority

---

*Corresponding Author. Email: jinchengqyh@126.com

classes will approach each other when the imbalance level goes high. Thrampoulidis et al. [23] theoretically study the Unconstrained-features SVM problem, whose global minima take the form of Simplex-Encoded-Labels Interpolation (SELI), a more general structure compared to the ETF. Dang et al. [20] inspects the ReLU-activated output features of deep linear network collapse to a general orthogonal frame for imbalanced data, where the orthonormal vectors of the frame are rescaled.

There is a line of works that connect the NC to the DNN performance [10, 24, 25]. Some researchers treat NC as a tool to alleviate minority collapse problem in imbalanced learning [26, 27, 28]. We reproduce these methods with backbone network ResNet50 [29] and datasets cifar10/cifar100, **Table** 3 shows NC-inspired methods ETF and ARB only outperform the plain model (ResNet50 with CE loss) slightly (or even worse in a few settings), which indicates that minority collapse is one but not the only problem that harms the generalization of learning model. One possibly important issue is that NC on training sample does not necessarily imply the NC on the distribution, as pointed out by Hui et al. [30]. This inconsistency can lead to severe performance degeneration when the sample is too scarce to represent certain classes.

Neural Collapse implies "maximal separateness" between classes, which inspires some works to consider fixed classifiers in the training[31, 32, 33, 34, 35]. However, these methods do not display advantages over the classical imbalanced learning methods equipped with learnable classifiers.

In this paper, we study the connection between the optimal structure induced by neural collapse and its corresponding classification rule, and propose a MSE-type loss function that improves the imbalanced learning with fixed classifier. Specifically,

1. Through the analysis of hard-to-predict feature distribution (the features that are distributed randomly around the mean of the classifiers), we find that the classification accuracy is improved if the features from the minor classes align with more directions and those from major classes with less directions, which corresponds to a decision rule called the **Generalized Classification Rule** (GCR) discriminated from the Regular 1-Nearest Neighbor Classification Rule (RCR) induced by general NC in literature, and we also term these directions as the **centers** of the classes.

2. We design an MSE-type objective that describes the average distance between the centers and a given feature. We show in the theoretical framework of Unconstrained Feature Model (UFM) that, for balanced or imbalanced data and fixed or learnable classifiers, the output features collapse but skew from the classifiers at terminal phase of training (TPT), which is different from the original Neural Collapse phenomenon and is termed "**N**eural **C**ollapse to **M**ultiple **C**enters" (NCMC) (Theorem 3.3 and 3.4); moreover, we find that RCR (with respect to classifiers) becomes a surrogate of GCR at NCMC (Remark 3.9 and Proposition C.1).

3. We design a practical loss function for fixed classifiers and a class-aware strategy for determining the number of centers for each class. The loss induces the NCMC which is justified in theory and experiments and achieves comparable performance on several datasets with varying imbalance ratios to the classical imbalanced learning methods such as LDAM [36], KCL [37], ARBLoss [27], which indicates that NCMC can improve generalization in imbalanced learning.

## 2 Preliminaries

### 2.1 Neural Collapse

Consider a classification task with $K$ classes and $n_k$ training samples per class, i.e., overall $N := \sum_{k=1}^{K} n_k$ samples. DNN-based classifiers generally have the form

$$\psi_\Theta(\mathbf{x}) = \mathbf{W_0}\mathbf{h_\theta}(\mathbf{x}) + \mathbf{b} \tag{1}$$

where $\mathbf{h_\theta}(\cdot) : \mathbb{R}^D \to \mathbb{R}^d$ is the feature mapping ($d \geq K$), $\mathbf{W}_0 = [\mathbf{w}_1, \ldots, \mathbf{w}_K]^\top \in \mathbb{R}^{K \times d}$ with $\mathbf{w}_k \in \mathbb{R}^d$ the weight vector of class $k$, and $\mathbf{b} \in \mathbb{R}^K$ the bias of classifier, respectively. $\Theta = \{\mathbf{W}_0, \mathbf{b}, \boldsymbol{\theta}\}$ is the set of the trainable network parameters, which includes the parameters $\boldsymbol{\theta}$ of a nonlinear compositional feature mapping (e.g., $\mathbf{h_\theta}(\mathbf{x}) = \sigma(\mathbf{W}_L(\ldots\sigma(\mathbf{W}_2\sigma(\mathbf{W}_1\mathbf{x}))\ldots)$ where $\sigma(\cdot)$ is an element-wise nonlinear function). Let $[A]$ be the set $\{1, 2, \ldots, A\}$ for positive integer $A$.

The network parameters are optimized by minimizing an empirical risk

$$\min_{\Theta} \frac{1}{N} \sum_{k=1}^{K} \sum_{i_k=1}^{n_k} \mathcal{L}\left(\mathbf{W_0}\mathbf{h}_{\boldsymbol{\theta}}\left(\mathbf{x}_{k,i_k}\right) + \mathbf{b}, \mathbf{y}_k\right) + \mathcal{R}(\boldsymbol{\Theta}), \tag{2}$$

where $\mathcal{L}(\cdot, \cdot)$ is a loss function (e.g., cross-entropy or MSE) and $\mathcal{R}(\cdot)$ is a regularization term (e.g., squared $L_2$-norm). Let us denote the feature vector of the $i_k$-th training sample of the $k$-th class by $\mathbf{h}_{k,i_k}$ (i.e., $\mathbf{h}_{k,i_k} = \mathbf{h}_{\boldsymbol{\theta}}\left(\mathbf{x}_{k,i_k}\right)$ ), with $i_k \in [n_k]$. Denote the centralized mean of feature from class $k$ by $\overline{\mathbf{h}}_k := \mathbf{h}_k - \boldsymbol{\mu}_G$ where $\mathbf{h}_k := \frac{1}{n_k} \sum_{i_k=1}^{n_k} \mathbf{h}_{k,i_k}$ and $\boldsymbol{\mu}_G := \frac{1}{K} \sum_{k=1}^{K} \mathbf{h}_k$; let $\overline{\mathbf{H}} := [\overline{\mathbf{h}}_1, \overline{\mathbf{h}}_2, \ldots, \overline{\mathbf{h}}_K]$.

Recently noticed NC phenomenon [10] shows the weight vectors align with the class-mean features

$$\mathbf{W_0}\overline{\mathbf{H}} \propto \overline{\mathbf{H}}^{\top}\overline{\mathbf{H}} \propto \mathbf{W_0}\mathbf{W_0}^{\top} \propto \frac{K}{K-1}\left(\mathbf{I}_K - \frac{1}{K}\mathbf{1}_K\mathbf{1}_K^{\top}\right) \tag{3}$$

with

$$\mathbf{h}_{k,1} = \mathbf{h}_{k,2} = \ldots = \mathbf{h}_{k,n_k} \tag{4}$$

for all $k \in [K]$ at the terminal phase of training. where we use $\mathbf{I}_K$ to denote the $K \times K$ identity matrix, $\mathbf{1}_K$ to denote the all-ones vector of size $K \times 1$. The alignment may have alternative shapes for other problem settings such as $\mathbf{h}$ is ReLU-activated feature or the data class is imbalanced.

## 2.2 NC for Unconstrained Features Model with Regularized MSE Loss

To understand the emergence of symmetric structures, recent papers study the "unconstrained features model" (UFM), where the features $\{\mathbf{h}\}$ and $\mathbf{W}_0$ are treated as free variables. The rationality behind this simplification is the powerful expressivity of a trained neural network. Some use UFM to study the NC phenomenon under MSE loss.

Let $\mathbf{H} = [\mathbf{h}_{1,1}, \ldots, \mathbf{h}_{1,n_1}, \mathbf{h}_{2,1}, \ldots, \mathbf{h}_{K,n_K}] \in \mathbb{R}^{d \times N}$. In balanced case where $n_1 = n_2 = \ldots = n_K$, $\mathbf{H}$ is associated with the one-hot vectors matrix $\mathbf{Y} = \mathbf{I}_K \otimes \mathbf{1}_n^{\top} \in \mathbb{R}^{K \times Kn}$, where $\otimes$ denotes the Kronecker product. The optimization of the following problem

$$\min_{\mathbf{W_0},\mathbf{H},\mathbf{b}} \frac{1}{2N}\left\|\mathbf{W_0}\mathbf{H} + \mathbf{b}\mathbf{1}_N^{\top} - \mathbf{Y}\right\|_F^2 + \frac{\lambda_{\mathbf{W_0}}}{2}\|\mathbf{W_0}\|_F^2 + \frac{\lambda_H}{2}\|\mathbf{H}\|_F^2 + \frac{\lambda_b}{2}\|\mathbf{b}\|_2^2, \tag{5}$$

gives the NC to ETF, where $\lambda_{\mathbf{W_0}}$, $\lambda_H$, and $\lambda_b$ are positive regularization hyper-parameters and $\|\cdot\|_F$ denotes the Frobenius norm. A closely related model is the Bias-Free models

$$\min_{\mathbf{W_0},\mathbf{H}} \frac{1}{2Kn}\|\mathbf{W_0}\mathbf{H} - \mathbf{Y}\|_F^2 + \frac{\lambda_{\mathbf{W_0}}}{2}\|\mathbf{W_0}\|_F^2 + \frac{\lambda_H}{2}\|\mathbf{H}\|_F^2, \tag{6}$$

which proves to converge to an Orthogonal Frame [19]. In the next section, we study a bias-free variant of (6).

# 3 Main Results

In this section we show why the regular classification rule is not optimal, and propose the generalized classification rule and its surrogate losses, then offer a UFM analysis on the NC phenomenon under these losses. Since our paper focuses on bias-free model, we will simply call $\mathbf{W}_0$ the classifier.

## 3.1 Nearest-Neighbor Classification Rule Revisit: A Toy Example

Let $\mathcal{X} \subset \mathbb{R}^{d_0}$ ($d_0$ is the input dimension) be the underlying data population, and $Z := \mathbf{h}(\mathcal{X}) \in \mathbb{R}^d$ represent the feature population of the trained backbone network $\mathbf{h}$; Assume the trained classifier $\mathbf{w}_1, \mathbf{w}_2, \ldots, \mathbf{w}_K$ align with training samples drawn from $\mathcal{X}$ perfectly and thus achieve zero training error.

Additionally, assume the classifiers are orthonormal and equally normed. Meanwhile, let the classes of training data $\mathcal{C}_1, \mathcal{C}_2, \ldots \mathcal{C}_K$ be arranged such that the class sample sizes follow a descending order, i.e. $n_1 > n_2 > \ldots > n_K$.

According to NC, when the neural network arrives at the terminal phase of training, the regular classification rule (**RCR**) with respect to $\mathbf{W}_0$ is

$$c = \operatorname{argmax}_{k \in [K]} \{\mathbf{w}_k^\top z\}_{k=1}^K, \tag{7}$$

i.e., the 1-nearest neighbor decision rule.

Assume $Z \sim \alpha_1 P_1 + \alpha_2 P_2$, the mixture of the subpopulation $P_1$ which is correctly classified by the decision rule and the subpopulation $P_2$ that is classified at random, where $\alpha_1$ and $\alpha_2$ are the positive weights with $\alpha_1 + \alpha_2 = 1$.

In the analysis, let $P_2$ have the form $z = \mu + p$, with $p \sim \mathcal{N}(0, sI_d)$ and $\mu := \overline{w}$ where $s$ is a small positive number and $\overline{w} = \frac{1}{K} \sum_{k=1}^K \mathbf{w}_k$, which is termed a **Hard-To-Predict** feature distribution near the global mean of the classifier. For convenience we denote events $E_k := \{z \in \mathcal{C}_k | z \sim P_2\}$ for all $k \in [K]$, and note that $\sum_{k=1}^K P\{E_k\} = 1$

By virtue of the expressivity of the deep neural network, the population of the major classes will be more concentrated to the classifier than that of the minor classes, so that populations trained by major classes have less probability falling in $P_2$, i.e,

$$P\{E_1\} \leq P\{E_2\} \leq \dots P\{E_K\}. \tag{8}$$

Then by the gaussianity of $p$ and the orthogonality of the classifiers $\mathbf{w}_1, \mathbf{w}_2, \dots, \dots, \mathbf{w}_K$, the probabilities of correctly classifying data from $P_2$ are the same, that is,

$$P\{\mathbf{w}_k^\top z > \max_{k' \neq k} \mathbf{w}_{k'}^\top z | z \in E_k\} = P\{\mathbf{w}_k^\top p > \max_{k' \neq k} \mathbf{w}_{k'}^\top p | p \sim \mathcal{N}(0, sI_d)\} = \frac{1}{K}, \tag{9}$$

since $\mathbf{w}_k^\top z > \mathbf{w}_{k'}^\top z \iff \mathbf{w}_k^\top(\mu + p) > \mathbf{w}_{k'}^\top(\mu + p) \iff \mathbf{w}_k^\top p > \mathbf{w}_{k'}^\top p$ given $\mathbf{w}_k^\top \mu = \mathbf{w}_{k'}^\top \mu$. Therefore, by the total probability

$$P_{RCR} := P\{z \text{ is correctly classified } | z \sim P_2\} = \frac{1}{K} \sum_{k=1}^K P\{E_k\} = \frac{1}{K}. \tag{10}$$

This calculation inspires us to come up with a different classification rule that outperforms RCR on $P_2$. Indeed, we can assign the label $y = k$ to the set $\{\widetilde{\mathbf{w}}_j^{(k)}\}_{j=1}^{f_k}$ of vectors, for all $k \in [K]$; denote $F = \sum_{k=1}^K f_k$ and $\overline{w}_{ext} = \frac{1}{F} \sum_{k=1}^K \sum_{j=1}^{f_k} \widetilde{\mathbf{w}}_j^{(k)}$; also similar to the RCR, assume all these $F$ vectors are equally normed and mutually orthogonal. These vectors correspond to a **Generalized Classification Rule** (GCR)

$$c = \operatorname{argmax}_{k \in [K]} \{ \max_{j \in [f_k]} \langle \widetilde{\mathbf{w}}_j^{(k)}, z \rangle \}, \tag{11}$$

or equivalently, $f_k$-nearest neighbor classification rule, where number of nearest neighbors, $f_k$, of an example depends on which class it belongs to. Again by the properties of normal distribution and total probability formula, we obtain

$$P_{GCR} := P\{z \text{ is correctly classified} | z \sim P_2\} = \frac{1}{F} \sum_{k=1}^K f_k P\{E_k\}. \tag{12}$$

Recall the ascending sequence of $P\{E_1\} \dots P\{E_K\}$, so that any choice of $f_1 \leq f_2 \leq \dots \leq f_K$ gives higher correctly-classified probability for $z \sim P_2$. If we in addition consider pairwise comparison instead of the one-vs-all fashion in (9), we have for any pair $k \neq k'$

$$P\{\langle \widetilde{\mathbf{w}}_k, z \rangle > \langle \widetilde{\mathbf{w}}_{k'}, z \rangle \mid z \in E_k \cup E_{k'}\} = \frac{1}{2}, \tag{13}$$

for **RCR**, and

$$P\{\max_{j \in [f_k]} \langle \widetilde{\mathbf{w}}_j^{(k)}, z \rangle > \max_{j' \in [f_{k'}]} \langle \widetilde{\mathbf{w}}_{j'}^{(k')}, z \rangle \mid z \in E_k \cup E_{k'}\} = \frac{f_k}{f_k + f_{k'}} \tag{14}$$

for **GCR**. In this case, we require $f_k < f_{k'}$ for $k < k'$ for all pairs of $k, k' \in [K]$, or equivalantly, $f_1 < f_2 < \dots < f_K$. The analysis as a whole has yielded the following proposition.

**Proposition 3.1.** *assume (1) all the classifiers, $\{\mathbf{w}_k\}$ in RCR or $\{\tilde{\mathbf{w}}_j^{(k)}\}$ in GCR are orthonormal frames; (2) $Z \sim \alpha_1 P_1 + \alpha_2 P_2$, the mixture of the subpopulation $P_1$ which is correctly classified by the decision rule and the subpopulation $P_2$ where $\alpha_1$ and $\alpha_2$ are the positive weights with $\alpha_1 + \alpha_2 = 1$; (3) $P_2$ has the form $z = \mu + p$ or $z = \mu_{ext} + p$ depending on which classification rules used, with $p \sim \mathcal{N}(0, sI_d)$ where $s$ is a small positive number; (4) $P\{E_1\} \leq P\{E_2\} \leq \ldots P\{E_K\}$. Then $P_{GCR} \geq P_{RCR}$ for $f_1 \leq f_2 \cdots \leq f_K$.*

The **Hard-To-Predict features** in the proposition 3.1 are considered to be drawn randomly around the mean of the classifiers. Thus, we are motivated to use more orthogonal directions to classify the minor class. However, the GCR in the analysis does not apply to the practical training of the neural network easily. Indeed, the loss vanishes quickly when training directly with this rule. For this reason, we consider finding a surrogate loss that a) induces neural collapse to an orthogonal frame, and b) the classification rule at the neural collapse approximate GCR.

## 3.2 Center and Multi-Center Frame

We first define the "centers" that resemble $\tilde{\mathbf{w}}_j^{(k)}$'s in **GCR** (11) (in definition 3.2). Let $f, f_1, f_2, \ldots, f_K \in \mathbb{Z}_+$ be preset factors such that $fK = \sum_{k=1}^{K} f_k$, $N = \sum_{k=1}^{K} n_k$, and $S := \sum_{k=1}^{K} f_k n_k$ and $\theta \in [0, \frac{\pi}{2}]$ be angle constant. Let the linear classifier $\mathbf{W}_0$ satisfies

$$\mathbf{w}_k^\top \mathbf{w}_k > 0, \quad and \quad \mathbf{w}_k^\top \mathbf{w}_{k'} = 0 \text{ if } k \neq k'. \tag{15}$$

and the data features $\mathbf{H} = [\mathbf{h}_{1,1}, \ldots, \mathbf{h}_{1,n_1}, \mathbf{h}_{2,1}, \ldots, \mathbf{h}_{K,n_K}] \in \mathbb{R}^{d \times N}$.

**Definition 3.2** (Center of Class $k$). Let $d > (f+1)K$, $\mathbf{V} := \left[\mathbf{v}_1^{(1)}, \ldots, \mathbf{v}_{f_1}^{(1)}, \mathbf{v}_1^{(2)}, \ldots, \mathbf{v}_{f_K}^{(K)}\right]^\top$ is a matrix consisting of $fk$ rows of $d$-dimensional vectors $\mathbf{v}_j^{(k)\top}$'s and satisfies equality

$$[\mathbf{V}^\top | \mathbf{W_0}^\top][\mathbf{V}^\top | \mathbf{W_0}^\top]^\top = \tag{16}$$

$$\operatorname{diag}(\|\mathbf{w}_1\|^2 \mathbf{I}_{f_1}, \|\mathbf{w}_2\|^2 \mathbf{I}_{f_2}, \ldots, \|\mathbf{w}_K\|^2 \mathbf{I}_{f_K}, \|\mathbf{w}_1\|^2, \|\mathbf{w}_2\|^2, \ldots, \|\mathbf{w}_K\|^2) \tag{17}$$

where $[\cdot | \cdot]$ is the column augmentation of the matrix, and $\operatorname{diag}(\cdot, \ldots, \cdot)$ is the diagonalization of the block matrices. Then a **center** of class $k$ is defined as

$$\mathbf{w}_j^{(k)} := \mathbf{v}_j^{(k)} \cos \theta + \mathbf{w}_k \sin \theta, \; j \in [f_k]. \tag{18}$$

A **multi-center frame** is the matrix consists of $fK$ rows of $\mathbf{w}_j^{(k)\top}$, i.e.

$$\mathbf{W} = \left[\mathbf{w}_1^{(1)}, \ldots, \mathbf{w}_{f_1}^{(1)}, \mathbf{w}_1^{(2)}, \ldots, \mathbf{w}_{f_K}^{(K)}\right]^\top, \tag{19}$$

Let $\mathcal{C}$ denote the constraint of the tuple $(\mathbf{V}, \mathbf{W}_0)$ such that $[\mathbf{V}^\top | \mathbf{W_0}^\top]$ satisfies (16) and $\|\mathbf{w}_k\|^2 > 0$ are positive for all $k \in [K]$.

By the definition 3.2, $\mathbf{w}_1^{(k)} = \mathbf{w}_2^{(k)} = \ldots = \mathbf{w}_{f_k}^{(k)}$ for all $k \in [K]$, verbally, the centers of each class are equally-normed, and $\mathbf{v}_j^{(k)\top} \mathbf{w}_k' = 0$ for all tuple $(k, k', j) \in [K] \times [K] \times [f_k]$ with $k' \neq k$. Note $d \geq (f+1)K$ is a necessary condition for the existence of $(f+1)K$ mutually orthogonal d-dim vectors in equation (16). Figure 3 illustrates the centers of Class 1 and Class 2.

Let $\overline{\mathbf{w}}^{(k)} := \frac{1}{f_k} \sum_{j=1}^{f_k} \mathbf{w}_j^{(k)}$ be the mean of the centers of class $k$. Since all $\mathbf{w}_j^{(k)}, j \in [f_k]$ are equally normed and equi-angular for each $k$, we can denote $\alpha_k^* = \angle(\overline{\mathbf{w}}^{(k)}, \mathbf{w}_j^{(k)})$ and $\rho_k^* = \angle(\overline{\mathbf{w}}^{(k)}, \mathbf{w}_k)$ with no ambiguity. It is easy to check that $\cos \alpha_k^* := \sqrt{\frac{\cos^2 \theta + f_k \sin^2 \theta}{f_k}}$ and $\cos \rho_k^* := \frac{f_k \sin \theta}{\sqrt{f_k \cos^2 \theta + f_k^2 \sin^2 \theta}}$.

Then we define the bias-free regression loss for the UFM w.r.t the feature $\mathbf{h}_{k,i_k}$ and the multi-center frame by

$$\frac{1}{2S} \|\mathbf{W}\mathbf{h}_{k,i_k} - \mathbf{y}_{k,i_k}\|_2^2 := \underbrace{\frac{1}{2S} \sum_{j=1}^{f} (\mathbf{w}_j^{(k)\top} \mathbf{h}_{k,i_k} - 1)^2}_{\text{align with the centers of the class}} + \underbrace{\frac{1}{2S} \sum_{j=1}^{f} \sum_{k' \neq k}^{K} (\mathbf{w}_{j'}^{(k')\top} \mathbf{h}_{k,i_k})^2}_{\text{seperate with the centers of other classes}} \tag{20}$$

to measure the average extent to which a feature $\mathbf{h}_{k,i_k}$ collapses to its class centers $\mathbf{w}_j^{(k)}$'s while stays away from centers of other classes, where $\mathbf{y}_{k,i_k}$ is the "$f_k$-hot coding":

$$\mathbf{y}_{k,1} = \mathbf{y}_{k,2} \ldots = \mathbf{y}_{k,f_k} = [\underbrace{0,\ldots,0}_{\sum\limits_{m=1}^{k-1} f_m \text{ 0's}}, \underbrace{1,\ldots,1}_{f_k \text{ 1's}}, 0,\ldots,0]^\top.$$

Let $\mathbf{Y} := [\mathbf{y}_{1,1}, \mathbf{y}_{1,2}, ..., \mathbf{y}_{1,n_1}, \mathbf{y}_{2,1}, \ldots, \mathbf{y}_{K,n_K}] \in \mathbb{R}^{F \times N}$, the minimization of the regression loss over all features subject to $\mathcal{C}$ turns into our prototype optimization problem in this paper

$$\mathbf{P}: \min_{\mathbf{H},\mathbf{V},\mathbf{W}_0} \frac{1}{2S}\|\mathbf{WH} - \mathbf{Y}\|_F^2 + \frac{\lambda_{\mathbf{W}_0}}{2}\|\mathbf{W}_0\|_F^2 + \frac{\lambda_H}{2}\|\mathbf{H}\|_F^2 \quad s.t. \ (\mathbf{V}, \mathbf{W}_0) \in \mathcal{C}. \tag{21}$$

### 3.3 Neural Collapse to Multiple Centers (NCMC)

The following theorem (proved in appendix D) characterizes the global solutions of the optimization when the data is balanced.

**Theorem 3.3.** *Given $n_1 = n_2 = \ldots = n_K$ and $d \geq (f+1)K$. If $K\sqrt{n\lambda_H\lambda_{\mathbf{W}_0}} \leq \cos\alpha^*$, then any global minimizer $(\mathbf{V}, \mathbf{W}_0^*, \mathbf{H}^*)$ of P satisfies*

$$\mathbf{h}_{k,1}^* = \ldots = \mathbf{h}_{k,n_k}^* = \mathbf{h}_k^* \propto \overline{\mathbf{w}}^{(k)}, \ \forall k \in [K]. \tag{22}$$

$$\langle \mathbf{h}_{k'}^*, \mathbf{h}_k^*\rangle = 0, \ \mathbf{w}_j^{(k)*^\top}\mathbf{h}_k^* = \mathbf{w}_{j'}^{(k')*^\top}\mathbf{h}_{k'}^*, \ \forall j, j' \in [f_k], \ and \ k, k' \in [K] \tag{23}$$

$$\mathbf{w}^*{}_1^\top\mathbf{h}^*{}_1 = \ldots = \mathbf{w}_K^\top\mathbf{h}^*{}_K, \ and \ \|\mathbf{w}_1^*\|^2 = \ldots = \|\mathbf{w}_K^*\|^2 = \frac{-K\lambda_{\mathbf{W}_0} + \sqrt{\frac{\lambda_{\mathbf{W}_0}}{n\lambda_H}}\cos\alpha^*}{\frac{\lambda_{\mathbf{W}_0}}{n\lambda_H}\cos^2\alpha^*}, \tag{24}$$

$$\lambda_{\mathbf{W}_0}\|\mathbf{w}_k\|_2^2 = n\lambda_H\|\mathbf{h}_k\|_2^2, \ and \ \|\mathbf{h}_1^*\|_2 = \ldots = \|\mathbf{h}_K^*\|_2, \tag{25}$$

*or otherwise, the objective P is minimized by $(\mathbf{V}, \mathbf{W}_0^*, \mathbf{H}^*) = (\mathbf{0}, \mathbf{0}, \mathbf{0})$.*

For imbalanced data and non-identical expansion factors $f_k$, the following theorem shows the relationship between the optimal conditions and the parameters $f_k$, $\theta$, and $n_k$ (proved in appendix D).

**Theorem 3.4.** *If $\cos\alpha_k^* > \frac{S}{f_k}\sqrt{\frac{\lambda_{W_0}\lambda_H}{n_k}}$, for all $k \in [K]$, then the global optimizer $(\mathbf{V}, \mathbf{W}_0^*, \mathbf{H}^*)$ of P satisfies*

$$\mathbf{h}_{k,1}^* = \ldots = \mathbf{h}_{k,n_k}^* = \mathbf{h}_k^* \propto \overline{\mathbf{w}}^{(k)}, \ \forall k \in [K]. \tag{26}$$

$$\mathbf{w}_j^{(k)*^\top}\mathbf{h}_k^* = \mathbf{w}_{j'}^{(k)*^\top}\mathbf{h}_k^*, \ \forall j, j' \in [f_k], \ , \ and \ k \in [K] \tag{27}$$

$$\langle \mathbf{h}_{k'}^*, \mathbf{h}_k^*\rangle = 0, \forall \ k \neq k' \tag{28}$$

$$\|\mathbf{w}_k^*\|^2 = \frac{-\frac{S\lambda_{W_0}}{f_k n_k} + \sqrt{\frac{\lambda_{W_0}}{n_k\lambda_H}}\cos\alpha_k^*}{\frac{\lambda_{W_0}}{n_k\lambda_H}\cos^2\alpha_k^*} \ and \ \|\mathbf{h}_k^*\|^2 = \frac{\lambda_{W_0}}{n_k\lambda_H}\|\mathbf{w}_k^*\|^2, \tag{29}$$

*or otherwise the objective P is minimized by $(\mathbf{V}, \mathbf{W}_0^*, \mathbf{H}^*) = (\mathbf{0}, \mathbf{0}, \mathbf{0})$.*

Compared to the Theorem 3.3, Theorem 3.4 indicates that $\|\mathbf{h}_k\|$, $\|\mathbf{w}_k\|$, and the ratio between them depends on all the expansion factors $f_j$'s and the size of the class $n_j$. Both theorems show the features of class $k$ converge in the direction of $\overline{\mathbf{w}}^{(k)}$, the mean of the centers of class $k$. We term this type of collapse "Neural Collapse to Multiple Centers (NCMC)".

*Remark* 3.5. The optimality of $\mathbf{V}$ is controlled by the centers.

*Remark* 3.6. NCMC differs from UFM analyses in existing literature since $\mathbf{h}_k^*$ and $\mathbf{w}_k^*$ are not aligned at optimum, and the norm of the optimal classifier depends on the expansion factors of the classes $f_k$.

**Corollary 3.7** (Corollary of Theorem 3.4). *The optimality condition of $P$ as $V$ and $W_0$ are both fixed satisfies*

$$\mathbf{h}^*_{k,1} = \ldots = \mathbf{h}^*_{k,n_k} = \mathbf{h}^*_k \propto \overline{\mathbf{w}}^{(k)} \, , \, \forall k \in [K]. \tag{30}$$

$$\langle \mathbf{w}^{(k)}_{j'}, \mathbf{h}^*_k \rangle = \langle \mathbf{h}^*_{k'}, \mathbf{h}^*_k \rangle = 0 \; \forall k' \neq k \in [K], j' \in [f_{k'}] \tag{31}$$

$$\mathbf{w}^{(k)\top}_1 \mathbf{h}_k = \ldots = \mathbf{w}^{(k)\top}_{f_k} \mathbf{h}_k, \forall k \in [K] \tag{32}$$

$$\|\mathbf{h}^*_k\|_2 = \frac{f_k \cos \alpha^*}{f_k \cos^2 \alpha^* + \lambda_H S} \tag{33}$$

*Remark* 3.8. According to the proof of the corollary, the fixed classifier case does not require the condition w.r.t the lower bound of $\cos \alpha^*$. Moreover, it is also clear that although $\mathbf{h}^*_k$ aligns with $\overline{\mathbf{w}}^{(k)}$, the length of it relies on the value of $f_k$ nonlinearly. Indeed, from the Corollary,

$$\|\mathbf{h}^*_k\|_2 = \frac{f_k \cos \alpha^*}{f_k \cos^2 \alpha^* + \lambda_H S} = \frac{1}{\cos \alpha^*_k + \frac{\lambda_H S}{f_k \cos \alpha^*_k}}$$

has the global maximum $\cos \alpha^*_k = \sqrt{\frac{\lambda_H S}{f_k}}$ if $\lambda_H S < f_k$.

*Remark* 3.9. NCMC induces an approximate rule to **GCR** in the following two aspects:

**(1)** the centers are "almost orthogonal" to each other: two centers from different classes are orthogonal to each other. The angle between two enters from the same class is $\arccos(\frac{\langle \mathbf{w}^{(k)}_j, \mathbf{w}^{(k)}_{j'} \rangle}{\|\mathbf{w}_k\|^2}) = \arccos \sin^2 \theta$ for $j \neq j'$. as $\theta$ is small, the angle is close to $\frac{\pi}{2}$.

**(2)** Under NCMC of our problem setting, the **RCR** w.r.t. the $W_0$ (see (7)) can be considered a surrogate of **GCR** to some extent: if a hard-to-predict feature can be classified by **RCR** with a margin correctly, then it can be classified correctly by **GCR** with probability larger than $\frac{1}{2}$. We will discuss this more formally in the proposition C.1.

### 3.4 NCMC for Fixed Classifier

In the Theorem 3.3 and Theorem 3.4, we present the NC conditions for $P$. However, solving $P$ requires optimization of a scaled orthonormal frame on a non-euclidean manifold, which is computationally expensive for overparameterized models. We hope the classifier can be fixed while not losing its performance severely. We first analyze the fixed classifier via UFM (proof in the appendix D), and later in the next section we propose a practical loss function for the fixed classifier.

### 3.5 Class-Aware Strategy for Determining the Number of Centers

The proposition3.1 indicates the extra dimensions help improve the classification of "hard-to-predict" samples in the distribution, and basically the expansion factors should satisfy $f_1 < f_2 < \ldots < f_K$ when the class size decreases, i.e., $n_1 \geq n_2 \geq \ldots \geq n_K$. The principle of generating these $f_k$'s is twofold: *(1)*: $f_1 \geq 1$; *(2)*: the ratio of ascending $\{f_k\}$) shall approximates the ratio of descending of $\{n_k\}$, i.e., $\frac{f_k}{f_{k+1}} \approx \frac{n_{K-k}}{n_{K-k-1}}$ for all $k \in [K]$.

Concretely, we use the **Class-Aware Strategy** to generate the expansion factors $f_k$ when $f \geq 2$:

Step 1: Given the descending list $[n_1, n_2, \ldots, n_K]$ and scale $[n_1, n_2, \ldots, n_K]$ to $\left[ \frac{n_1}{N}, \frac{n_2}{N}, \ldots, \frac{n_K}{N} \right]$;

Step 2: Calculate $[a_1, a_2, \ldots, a_K]$ where $a_k = \lfloor \frac{(f-1)Kn_k}{N} \rfloor + 1$, to ensure each element is positive;

Step 3: Reverse the order of the list to $[a_K, a_{K-1}, \ldots, a_1]$ then add 1 from the left until the sum of the elements in the list equals $fK$.

For example, when $(n_1, n_2, n_3) = (1, 3, 3)$ and $f = 3$ then Step 1 outputs $(\frac{1}{7}, \frac{3}{7}, \frac{3}{7})$; Step 2 outputs $(1, 3, 3)$; Step 3 outputs $(f_1, f_2, f_3) = (4, 4, 1)$.

# 4 Experiments

In this section, we **(1)** propose a cosine loss function for fixed classifier; **(2)** verify NCMC induced by the cosine loss through experiments; **(3)** show how $f$ and $\theta$ influence the learning performance; **(4)** Compare long-tail classification performance to SETF method with fixed classifier and other classical methods with learnable classifier.

## 4.1 Datasets and Training Schedule

We set long-tailed classification tasks on five datasets, CIFAR-10 [38], CIFAR-100 [38], SVHN [39], STL-10 [40], and large dataset ImageNet [41]with two architectures ResNet-50 and densnet-150 (details in appendix F). Let $\tau := \frac{n_{max}}{n_{min}}$ represent the imbalance ratio of the long-tailed sampling from the dataset, where $n_{max}$ and $n_{min}$ are the size of the largest class and the smallest class, resp. The accuracy results are the average of three repeated experiments with different seeds. The best and second-best results are boldfaced and underlined.

## 4.2 The Cosine Regression Loss

**The Cosine Regression Loss.** Motivated by the toy example 3.1 and the theoretical justification of NCMC, we propose the regularized loss for the fixed unit-norm multi-center frame that satisfies definition 3.2:

$$\mathcal{L}(\mathbf{W}, \mathbf{h}_{k,i_k}) = \beta \sum_{j=1}^{f_k} \mathrm{Cos}(\mathbf{w}_j^{(k)}, \mathbf{h}_{k,i_k}) + \lambda \left( \|\mathbf{h}_{k,i_k}\| - 1 \right)^2. \tag{34}$$

where $\mathrm{Cos}(\mathbf{w}, \mathbf{h}) = \|\langle \mathbf{w}, \frac{\mathbf{h}}{\|\mathbf{h}\|} \rangle - 1\|_2^2$ is termed **Cosine Loss**, $\lambda$ and $\beta$ are regularization coefficients (we relegate the selection of the coefficients to appendix F ). This is called a Cosine loss discard all $\mathbf{w}_j^{(k')}$ terms of loss (20) for features from class $k$, where $k' \neq k$, since in practice $h_k$ aligns with $\overline{\mathbf{w}}^{(k)}$ only if $\langle \mathbf{w}_j^{(k')}, \mathbf{h_k} \rangle = 0$. The derivative of the loss with respect to some feature $\mathbf{h}$ then is given by

$$\frac{\mathrm{d}\,\mathrm{Cos}(\mathbf{w}, \mathbf{h})}{\mathrm{d}\mathbf{h}} = -\frac{2}{\|\mathbf{h}\|^2} \cdot (1 - \frac{\langle \mathbf{w}, \mathbf{h} \rangle}{\|\mathbf{h}\|}) \cdot (\mathbf{w} - \frac{\langle \mathbf{w}, \mathbf{h} \rangle \mathbf{h}}{\|\mathbf{h}\|}), \tag{35}$$

(the derivation is postponed to the Appendix E). It shows the gradient changes both the magnitude and the direction of the features with a scale $\|\mathbf{h}\|^{-2}$ of the feature, compared to the dot-regression loss in [26]. The regularizer guarantees the gradient does not vanish for the features with large norm and explode for the features with small norm since $\mathrm{Cos}(\mathbf{w}, \mathbf{h})$ is scaled by $\|\mathbf{h}\|^{-2}$.

Note that, the loss uses the Multi-Center Frame $W$ in the training while, due to Remark 3.9, we still use $\mathbf{W}_0$ as the classifier. In the context, we term the objective (34) as $f_1 = f_2 = \ldots = f_K$ the "Average Loss" (AL), and that with $f_k$ selected by class-aware strategy the "Class-Aware Loss" (CAL).

## 4.3 Neural Collapse

We design experiments to verify loss **P**, AL and CAL induce NCMC. The collapse is measured by $\mathcal{NC} := \left\| \mathbf{h}_{k,i_k} / \|\mathbf{h}_{k,i_k}\| - \overline{\mathbf{w}}^{(k)} / \left\| \overline{\mathbf{w}}^{(k)} \right\| \right\|_2^2$. For simplicity, we calculate the mean and standard deviations of the vector $\mathbf{h}_{k,i_k} / \|\mathbf{h}_{k,i_k}\| - \overline{\mathbf{w}}^{(k)} / \left\| \overline{\mathbf{w}}^{(k)} \right\|$ when the mean and standard deviation tends to zero, we can show that NC occurs. We fix $f = 20$ and $\theta = 0.2$; we pick the tuple $[f_1, f_2, \ldots, f_K]$ by the Class-Aware Strategy. Figure 1 shows that both AL and CAL induce NC. The change of mean is rapid and the standard deviation first increases and then converges to zero slower. Variability is defined as the average of the sum of $\sigma = 1 - \frac{\mathbf{h}_{k,i_k}}{\|\mathbf{h}_{k,i_k}\|} \frac{\mathbf{w}_k}{\|\mathbf{w}_k\|}$ over all $k$ and $i_k$, and measures the alignment of $\mathbf{H}$ and $\mathbf{W}$, and we observe the in all three objectives the variabilities stay away from zero. Figure 4 plots the NCMC of loss **P** with or without regularization on the norm of features. The regularization results in heavier NC.

We also provide plots of NCMC on VGG and LeNet for CAL that demonstrate the universality of the phenomenon (refer to **Table** 5). We also draw heatmaps of the neural collapse for better visualization. Fig.6 is the heatmap of $\tilde{\mathbf{H}}^\top \tilde{\mathbf{H}}$ where $\tilde{H} = [\tilde{\mathbf{h}}_1, \tilde{\mathbf{h}}_2, \ldots \tilde{\mathbf{h}}_K]$ and $\tilde{\mathbf{h}}_k := \frac{\mathbf{h}_k}{\|\mathbf{h}_k\|}$ is the normalized class-mean features. Fig.7 is the heatmap for the normalized features in class 9 of CIFAR-10; it is noted that all features in this class stay close to each other from the initialization to the end of training.

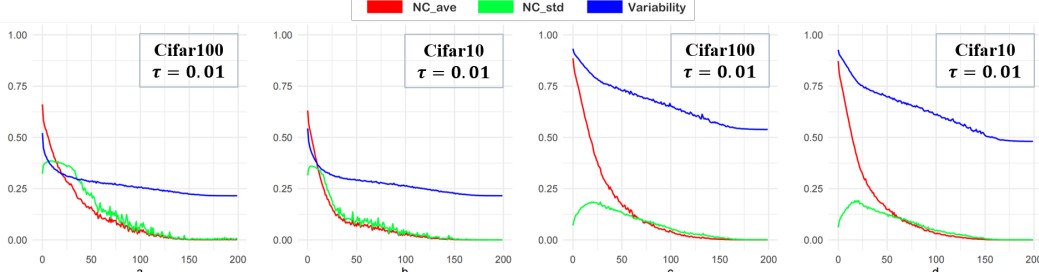

Figure 1: An illustration of the Neural Collapse curves of AL (a and b, on CIFAR100 with $\tau = 0.01$ and CIFAR 10 with $\tau = 0.01$ respectively), CAL (c and d, on CIFAR100 with $\tau = 0.01$ and CIFAR 10 with $\tau = 0.01$ respectively). The AL is equipped with f=20 and CAL uses Class-Aware Strategy.

## 4.4 Long-Tailed Classification

We conduct an ablation study with ResNet50 on CIFAR-100. **Table** 1 presents the performance of ResNet50 on both balanced CIFAR-100 and its imbalanced sample with or without Mixup training under CAL at imbalance ratio $\tau = \{0.005, 0.01, 0.02, 1(balanced\ case)\}$. We do not use objective **P** since it is inferior to the Cosine Regression Loss (see **Table** F.1). **P** is incompatible with mixup training and it may be degenerated by the myriad of distraction terms in the objective.

As shown in **Table** 1, the performances get improved for almost all imbalanced ratios when the CE is replaced with our designs, especially CAL. And our method is compatible with Mixup training. However, the SETF and our method have lower accuracy than the CE in the balanced case. The analysis in the Proposition 3.1 tells us that in the balanced case where $f_k$'s are identical, our theory will not have a significant positive effect on the classification. Our method has a similar form to SETF

Table 1: An ablation study of Mixup method for training ResNet50 on CIFAR-100 using different classifiers and loss functions. The numbers in the second row are imbalance ratios. The parameters are fixed with $f = 20$, and $\theta = 0.2$

| Methods | without Mixup | | | | with Mixup | | | |
|---|---|---|---|---|---|---|---|---|
| | 0.005 | 0.01 | 0.02 | balanced | 0.005 | 0.01 | 0.02 | balanced |
| **ResNet50** | | | | | | | | |
| **CE** | 35.6±0.3 | 37.5±0.4 | 43.3±0.2 | **79.5**±0.3 | 42.4±0.5 | 46.7±0.3 | 52.7±0.3 | **81.8**±0.1 |
| **SETF** | 38.1±0.4 | 42.6±0.2 | 48.4±0.3 | 78.7±0.2 | 43.0±0.1 | 48.3±0.5 | 52.5±0.3 | 79.7±0.2 |
| **CAL** | **40.6**±0.3 | **44.7**±0.2 | **50.2**±0.5 | 78.5±0.4 | **46.5**±0.5 | **50.1**± 0.3 | **54.3**±0.4 | 79.4±0.4 |

who fixes the ETF classifier and emphasizes the gradient norm balance among the classes. On the contrary, CAL concentrates on the fitting to the multiple centers. We compare our method CAL to CE and SETF on four small datasets CIFAR-10, CIFAR-100, SVHN, STL-10 (**Table** 2, and **Table** 5). It shows the stability of our Class-Aware Strategy and displays a significant improvement to the original networks.

The hyper-parameters have influences on the performance: if $\theta$ is small, the centers of a class are very close to each other, else if $\theta$ gets larger, the margin of the classifier will be smaller; parameter $f$ encodes partial imbalance information of the classes, when $f$ is small all $f_k$ are close to each other and cannot offer useful supervision for the "hart-to-predict" samples. Refer to appendix F for more information about hyperparameter selection.

We summarize in **Table** 6 the performance of CAL as $f$ and $\theta$ vary. The result empirically demonstrates: **(1)** fixing $f$, the accuracy roughly peaks at some $\theta$ bounded away from 0 and $\pi/2$. **(2)** at $\theta = \pi/2$, The frame collapses to the classifier, resulting in an accuracy rate similar to SETF [26] where the norm of the gradient is weighted by the class imbalance ratio; **(3)** Frame that is orthogonal

Table 2: Long-tailed classification accuracy (%) with ResNet and DenseNet on CIFAR-10 and CIFAR-100.

| Methods | CIFAR-10 | | | CIFAR-100 | | |
|---|---|---|---|---|---|---|
| | 0.005 | 0.01 | 0.02 | 0.005 | 0.01 | 0.02 |
| *ResNet* | | | | | | |
| CE | 72.3±0.1 | 78.6±0.2 | 84.0±0.1 | 42.4±0.5 | 46.7±0.3 | 52.7±0.3 |
| SETF | 74.2±0.5 | 79.7±0.4 | 83.8±0.3 | 43.0±0.1 | 48.3±0.5 | 52.5±0.3 |
| CAL | **80.0**±0.5 | **84.1**±0.4 | **85.9**±0.2 | **46.5**±0.5 | **50.1**± 0.3 | **54.3**±0.4 |
| *DenseNet* | | | | | | |
| **CE** | 71.1±0.5 | 77.7±0.3 | 84.1±0.1 | 42.9±0.2 | 47.4±0.2 | 53.3±0.2 |
| **SETF** | 72.9±0.4 | 78.5±0.3 | 83.4±0.3 | 42.3±0.2 | 46.3±0.3 | 52.6±0.2 |
| **CAL** | **78.1**±0.2 | **81.1**±0.2 | **84.5**±0.2 | **46.3**±0.3 | **50.1**±0.2 | **54.0**±0.2 |

to the classifier does not learn anything in the training since the representation is irrelevant to the classifier, thus providing no useful discrimination information. We compare our method to several

Table 3: A comparison of several recent methods of long-tail classification trained on ResNet50. $f = 20$ and $\theta = 0.2$ are fixed. The values without $\pm$ are that we did not reproduce.

| Methods | Cifar-10 | | | | Cifar-100 | | | | ImageNet |
|---|---|---|---|---|---|---|---|---|---|
| | 0.005 | 0.01 | 0.02 | 0.1 | 0.005 | 0.01 | 0.02 | 0.1 | |
| **CE** (Mixup) | 72.3±0.1 | 78.6±0.2 | 84.0±0.1 | 91.9±0.1 | 42.4±0.5 | 46.7±0.3 | 52.7±0.3 | **67.9**±0.1 | 44.2±0.3 |
| **LDAM-DRW** | 74.6±0.3 | 80.1±0.3 | 84.1±0.2 | 90.0±0.2 | 39.5±0.3 | 44.2±0.2 | 50.0±0.2 | 62.5±0.2 | 47.7 |
| **KCL** | 75.0±0.3 | 80.9±0.2 | 84.5±0.3 | 90.7±0.4 | 40.3±0.4 | 44.8±0.3 | 50.2±0.2 | 63.0±0.2 | 51.5 |
| **SETF** | 74.2±0.5 | 79.7±0.4 | 83.8±0.3 | 91.3±0.4 | 43.0±0.1 | 48.3±0.5 | 52.5±0.3 | 66.1±0.3 | 44.7 |
| **ARBloss** | 79.5±0.7 | 83.8±0.4 | **86.1**±0.3 | 91.5±0.3 | 42.7±0.8 | 47.1±0.5 | 49.7±0.2 | 64.4±0.4 | **52.8** |
| **CAL** | **80.0**±0.5 | **84.1**±0.4 | 85.9 ±0.2 | **92.0**±0.3 | **46.5**±0.5 | **50.1**±0.3 | **54.3**±0.4 | 65.9±0.3 | 49.7±0.2 |

classical methods including NC-inspired methods ARB loss and SETF, margin-based LDAM-DRW, contrastive learning method KCL and original CE. The **Table** 3 shows the methods in comparison. We observe that our method is comparable to or even better than the others. The comparison indicates: **(1)** Our method has some advantages for heavily imbalanced cases. One of the underlying mechanisms is when the minor classes are underestimated due to the lack of sample, they are likely to display the gaussianity such that the proposition 3.1 and class-aware strategy work fine; as the imbalance ratio $\tau$ rise, the minor classes lose their randomness during training, where our method fails. **(2)** Our method does not compete with the KCL and ARBLoss on ImageNet, reflecting the limitations of our method in the flexibility of the direction and magnitude of the classifier weights when learning large datasets. We also compare our method to a recent work RBL [42] in the Appendix H that demonstrates the potential limitation of the fixed classifier and the advantage of CAL. **(3)** two classical methods LDAM-DRW and KCL do not compete with other NC-inspired methods trained with Mixup training, demonstrating the effectiveness Mixup training strategy. **(4)** Our method with the parameter chosen in the experiment outperforms CE negligibly or performs worse than CE when the imbalance ratio approaches 1. However, we find picking $f = 10$, $\theta = 0.5$ gives an accuracy rate **92.5** $\pm$ 0.2 on Cifar-10 with $\tau = 0.1$, which shows the significance of hyperparameter selection for our method.

## 5 Conclusion

In this paper, we rethink the regular 1-Nearest Neighbor Classification Rule (RCR) in imbalanced learning; an analysis of the Hard-To-Predict feature indicates under certain circumstances the generalized classification rule (GCR) is superior to RCR, which implies that minor classes should compare to more "neighbors" in the classification. Then we introduce neural collapse to multiple centers (NCMC) under an MSE-type loss, where the centers play a role similar to the neighbors in GCR. According to the framework of the Unconstrained Features Model, the features of each class collapse to the class mean of the centers in balanced or imbalanced settings for learnable or fixed classifiers. We notice that at NCMC, RCR resembles GCR in terms of the hard-to-predict feature distribution. We then propose the cosine loss, a surrogate regression objective of the MSE-type loss called Cosine Loss, that applies to the fixed classifier; and develop the class-aware strategy for determining the number of centers of each class, inspired by the analysis of the Hard-To-Predict Feature. The cosine loss practically induces the NCMC at the terminal phase of training; the combo of the class-aware strategy and the loss with the fixed classifier demonstrates its effectiveness in long-tailed classification. Our work shows the possibility of obtaining a task-specific classification rule by designing the optimal structure at neural collapse under customized losses; it provides a connection among the optimal structure of the feature-classifier alignment, the classification rule, and the generalization in the learning problem.

## Acknowledgement

This work was supported by the National Natural Science Foundation of China (Nos. 62136005, 62276162, 62476160, 62306170), the National Science and Technology Major Project (No.2021ZD0112400), the Science and Technology Major Project of Shanxi (No. 202201020101006), the Special Fund for Science and Technology Innovation Teams of Shanxi Province (No. 202304051001001)

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

# A    Related Work

## A.1    Neural Collapse

Since the original paper [10], neural collapse has been intensively studied, with [12, 11] introducing the now widely accepted unconstrained features model (UFM), and layer peeled model. The literature of NC can be loosely categorized into: *(a)* the study of the geometry of NC [43, 44, 45, 11, 46, 47, 12, 17, 33, 48, 49, 50], *(b)* finding connection between NC phenomena and NN performance [27, 25, 51, 52, 26, 53, 54, 35, 55] and *(c)* empirical observations concerning NC [56, 57].

## A.2    Long-Tailed Learning

Long-tail learning refers to the training from data with long tailed-class distributions [58]. We list several categories of methods that are related to our work. **Re-sampling.** Early works [59, 60, 61, 62] re-sampling is a sampling strategy that balances the sample used for learning by over-sampling the samples from minor classes or under-sampling the samples from major classes. Over-sampling may make the model overfit these classes and harm the generality. Under-sampling removes some samples of major classes, which might remove the key data for representation learning and bring the performance drop.

**Re-weighting.** Another idea is to assign different weights for different classes, even instances. [63, 64, 65, 66] re-weight the loss according to the sample size of the class. [67] introduce the prior probabilities from a Bayesian view and balance the exponential logits via the class sample size. Focal loss [68, 69] re-weight each instance according to their hard-to-learn level, i.e., making the model take more care of the wrong-recognized samples. Wang et al. [70] provide a data-dependent generalization bound that explains the success of re-weighting and logit adjustment strategy.

**Two-stage Learning.** The representations learned from instance-balancing sampling are believed to be general. Many researchers attempt to separate learning process to two stages. [36] trains the model in a normal approach in stage 1 and then uses deferred resampling strategy to fine-tune with class-balanced resampling or uses deferred re-weighting to re-weight different classes in stage 2. [71] disentangles the basic feature learning and re-balancing learning via two separated branches.[72, 73] decouple the representation learning and classification. They claim that instance-balanced sampling gives more general representation and originally use instance-balanced sampling to learn the representations at first stage, then fix the feature and retrained the classifier (cRT) by techniques such as label-aware smoothing (LAS) and learnable weight scaling (LWS) in the second.

**Contrastive Learning.** Contrastive learning is a learning paradigm that learns representation that maximizes the similarity between positive and negative samples [74]. Khosla et al. [75] initiated Supervised Contrastive Learning (SCL) paradigm by leveraging class labels. For imbalanced classification tasks, Contrastive learning usually faces imbalanced positive/negative pairs of samples. To balance the feature space, KCL [37] balances feature space by using the same number of positive pairs for all the classes; class complement methods [76, 77] are proposed to construct positive and negative pairs for the rebalance of the class. Recent advances in multi-modal foundation models such as CLIP [78] and VLLTR [79] have displayed remarkable performance on the downstream long-tail classification tasks.

**Learning to Mitigate Minority Collapse.** Yang et al. [26] propose the method (we name it "SETF") that learns the representation from imbalanced data with last-layer classifiers fixed as a simplex ETF during training and prove the optimal last-layer features converge to ETF structure; Gao et al. [42] propose the method that optimize the ETF structure under rotation and the uses post-hoc logit-adjustment for prediction. Liu et al. [28] use NC Regularization to minimize the within-class variability and maximize the between-class separateness of the output features; Inspired by neural collapse phenomenon in balanced case, Xie et al. [27] design class-balanced CE loss (termed "ARB loss") that aims to balance the gradient among classes. Zhong et al.[54] discover the minority collapse phenomenon in semantic segmentation and use neural collapse as a regularization to improve the discriminate ability of the network. **Learning Test Agonistic Label distribution.** In [80], the authors consider an imbalanced learning task where the training data is long-tailed while the distribution shift between training data and test data is unknown. They deal with the problem by applying diverse

expert networks in training to handle different class distributions and aggregate the experts in the test time. Yang et al. [81] propose another expert-mixing strategy that tackles the mild changes in the label distributions.

**Learning with fixed classifiers.** Apart from SETF, other symmetric structures, such as regular polytopes [31, 32, 33], Hadamard matrix [34], and hierarchy-aware frame [35] have the classifier fixed in the network training process. Additionally, fixing the classifier is also a computationally friendly strategy since the classifier does not need backpropagation.

# B   Illustrations

## B.1   A Toy Example of Hard-to-Predict Sample

Fig 2 is an illustration of a Hard-To-Predict sample (the purple dots). The points are simulated from the 3-dim multivariate normal distributions, where colors indicate different covariances and means. The red, green, and blue represent the unseen sample from the distribution (Call it $P_1$) that can be well-classified by the trained model. The purple points are drawn from the distribution ($P_2$ in the proposition) randomly classified by the trained model. The whole feature space of the underlying data distribution is assumed to be $P = \alpha_1 P_1 + \alpha_2 P_2$.

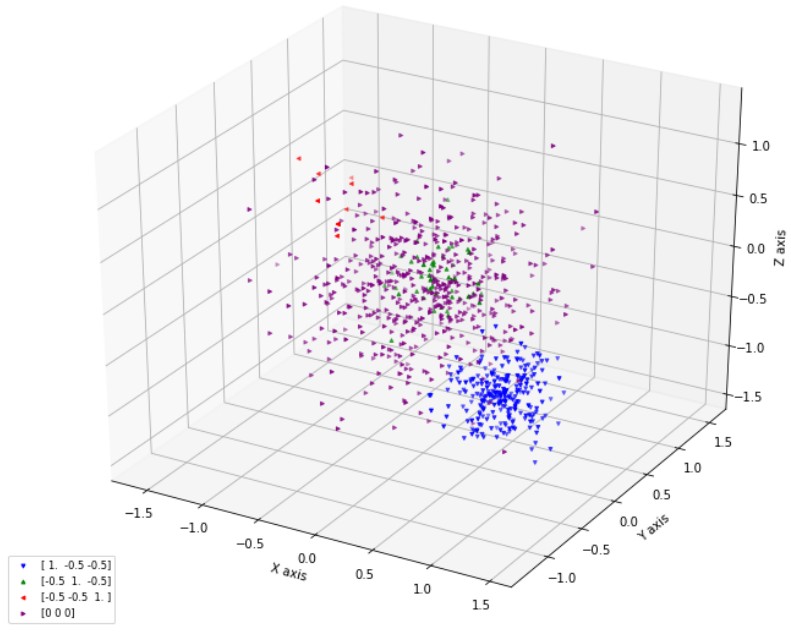

Figure 2: An illustration of the toy example

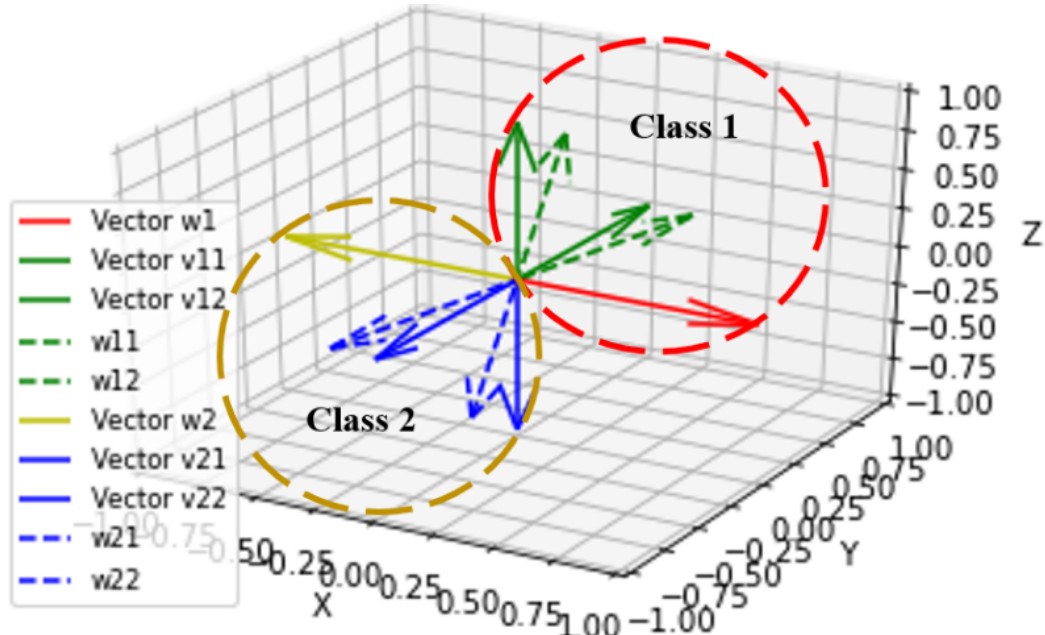

Figure 3: The 3D illustration of multiple centers for class 1 and class 2. The dashed blue vectors are the two centers of class 1, and the dashed green are the two centers of class 2. And $w_{ij} = \cos\theta * v_{ij} + \sin\theta * w_i$, where $i, j \in \{1, 2\}$.

### B.2 Multiple Centers of Two Classes

## C Regular Classification Rule v.s. Generalized Classification Rule at NCMC

In the following proposition C.1, let $z$ be the hard-to-predict feature sampled from $P_2$ at NCMC, i.e. $z = \mu' + p$ where $p \sim \mathcal{N}(0, s\,\mathrm{I}_d)$, and $\mu' = \frac{1}{K}\sum_{k=1}^{K} \overline{\mathbf{w}}^{(k)}$. Denote $\gamma := \max_{j\in[f_k]}\langle \mathbf{v}_j^{(k)}, z\rangle - \max_{j'\in[f_{k'}]}\langle \mathbf{v}_{j'}^{(k')}, z\rangle$. The difference here to proposition 3.1 is the mean $\mu'$ induced by NCMC, instead of original NC without the centers under the regular MSE loss. Then we prove that in one-vs-one setting, the original RCR

$$c = \underset{k\in[K]}{\operatorname{argmax}}\{\mathbf{w}_k^\top z\}. \tag{36}$$

with a small margin implies correct classification with probability over $\frac{1}{2}$ by GCR.

**Proposition C.1.** *Let* $\mathbf{W}$ *be defined as in definition 3.2 with unit-norm centers and classifiers; assume* $f_k > f_{k'}$. *Then there exists an* $\epsilon_0 > \frac{\cos^2\theta}{f_k K} - \frac{\cos^2\theta}{f_{k'} K}$ *such that*

*(1) for any* $\epsilon < \epsilon_0$, $\langle \mathbf{w}_k, z\rangle - \langle \mathbf{w}_{k'}, z\rangle > \frac{\cos^2\theta}{K\sin\theta}\left(\frac{1}{f_{k'}} - \frac{1}{f_k}\right) - \frac{\epsilon}{\sin\theta}$ *implies*

$$P\{\max_{j\in[f_k]}\langle \mathbf{w}_j^{(k)}, z\rangle > \max_{j'\in[f_{k'}]}\langle \mathbf{w}_{j'}^{(k')}, z\rangle\} > \frac{1}{2};$$

*(2) for any* $\epsilon' > \epsilon_0$, $\langle \mathbf{w}_k, z\rangle - \langle \mathbf{w}_{k'}, z\rangle < \frac{\cos^2\theta}{K\sin\theta}\left(\frac{1}{f_{k'}} - \frac{1}{f_k}\right) - \frac{\epsilon'}{\sin\theta}$ *implies*

$$P\{\max_{j\in[f_k]}\langle \mathbf{w}_j^{(k)}, z\rangle < \max_{j'\in[f_{k'}]}\langle \mathbf{w}_{j'}^{(k')}, z\rangle\} > \frac{1}{2}.$$

*Proof.* Due to the orthogonality of $\mathbf{V}$ and $\mathbf{W}_0$, $\langle \mathbf{v}_j^{(k)}, z\rangle$ or $\langle \mathbf{v}_{j'}^{(k')}, z\rangle$ are still independent isotropic Gaussian, conditioning on the value of

$$\langle \mathbf{w}_k, z\rangle - \langle \mathbf{w}_{k'}, z\rangle = \langle \mathbf{w}_k, p\rangle - \langle \mathbf{w}_{k'}, p\rangle \tag{37}$$

since

$$\mathbf{w}_k \mu' = \mathbf{w}_{k'} \mu'. \tag{38}$$

Moreover, the maximum of i.i.d gaussians has continuous (differentiable) density, so $\ell(x) := P\{\gamma > x\}$ is a continuous decreasing function, where $\ell(-\infty) = 1$, $\ell(\infty) = 0$, and

$$\ell(\frac{\cos^2 \theta}{f_k K} - \frac{\cos^2 \theta}{f_{k'} K}) = P\{\max_{j \in [f_k]} \langle \mathbf{v}_j^{(k)}, p \rangle > \max_{j' \in [f_{k'}]} \langle \mathbf{v}_{j'}^{(k')}, p \rangle\} = \frac{f_k}{f_k + f_{k'}},$$

Therefore there exists only one $\epsilon_0 > \frac{\cos^2 \theta}{f_k K} - \frac{\cos^2 \theta}{f_{k'} K}$, such that,

$$\ell(\frac{\epsilon_0}{\cos \theta}) = \frac{1}{2}.$$

and

$$\ell(\frac{\epsilon}{\cos \theta}) > \frac{1}{2} \quad for \ \epsilon < \epsilon_0. \tag{39}$$

$$\ell(\frac{\epsilon'}{\cos \theta}) < \frac{1}{2} \quad for \ \epsilon' > \epsilon_0. \tag{40}$$

Then with probability over $\frac{1}{2}$,

(1) the last few conditions implies

$$\max_{j \in [f_k]} \langle \mathbf{w}_j^{(k)}, z \rangle - \max_{j' \in [f_{k'}]} \langle \mathbf{w}_{j'}^{(k')}, z \rangle \tag{41}$$

$$= \max_{j \in [f_k]} \langle \mathbf{v}_j^{(k)} \cos \theta, z \rangle + \langle \mathbf{w}_k \sin \theta, z \rangle - \max_{j \in [f_k]} \langle \mathbf{v}_{j'}^{(k')} \cos \theta, z \rangle - \langle \mathbf{w}_{k'} \sin \theta, z \rangle \tag{42}$$

$$= \gamma \cos \theta + \sin \theta (\langle \mathbf{w}_k, p \rangle - \langle \mathbf{w}_{k'}, p \rangle) + \frac{\cos^2 \theta}{f_k K} - \frac{\cos^2 \theta}{f_{k'} K} \tag{43}$$

$$\geq 0 \tag{44}$$

for $\epsilon < \epsilon_0$.

(2) Similar to (1), using formula (40) and (38), we deduce that with probability over $\frac{1}{2}$:

$$\max_{j \in [f_k]} \langle \mathbf{w}_j^{(k)}, z \rangle - \max_{j' \in [f_{k'}]} \langle \mathbf{w}_{j'}^{(k')}, z \rangle \tag{45}$$

$$= \gamma \cos \theta + \sin \theta (\langle \mathbf{w}_k, p \rangle - \langle \mathbf{w}_{k'}, p \rangle) + \frac{\cos^2 \theta}{f_k K} - \frac{\cos^2 \theta}{f_{k'} K} \tag{46}$$

$$\leq 0 \tag{47}$$

for $\epsilon' > \epsilon_0$. $\qquad\qquad\qquad\qquad\qquad\qquad\qquad\qquad\qquad\qquad\qquad\qquad\qquad\qquad\qquad\qquad\square$

## D  Proof of the Theorems and Their Corollaries

*Proof of Theorem 3.3.* In this proof we consider the balanced case where $n_1 = n_2 = \ldots = n_K$, and $S = fN$, $\mathbf{Y} = \mathbb{I}_K \otimes 1_f \otimes 1_n^\top$. The proof is to lower bound the

$$f(\mathbf{W}, \mathbf{H}) := \frac{1}{2fN} \|\mathbf{WH} - \mathbf{Y}\|_F^2 + \frac{\lambda_{\mathbf{W}_0}}{2} \|\mathbf{W_0}\|_F^2 + \frac{\lambda_H}{2} \|\mathbf{H}\|_F^2$$

by a column of inequalities. First, observe that

$$\frac{1}{2fN}\|\mathbf{W}\mathbf{H} - \mathbf{Y}\|_F^2 + \frac{\lambda_{\mathbf{W}_0}}{2}\|\mathbf{W_0}\|_F^2 + \frac{\lambda_H}{2}\|\mathbf{H}\|_F^2 \tag{48}$$

$$= \frac{1}{2Kfn}\sum_{k=1}^{K}\sum_{i=1}^{n}\|\mathbf{W}\mathbf{h}_{k,i} - \mathbf{y}_k\|_2^2 + \frac{\lambda_{\mathbf{W}_0}}{2}\sum_{k=1}^{K}\|\mathbf{w}_k\|_2^2 + \frac{\lambda_H}{2}\sum_{k=1}^{K}\sum_{i=1}^{n}\|\mathbf{h}_{k,i}\|_2^2 \tag{49}$$

$$= \frac{1}{2Kfn}\sum_{k=1}^{K}\sum_{j=1}^{f}\sum_{i=1}^{n}\left(\mathbf{w}_j^{(k)^\top}\mathbf{h}_{k,i} - 1\right)^2 + \frac{1}{2Kfn}\sum_{k=1}^{K}\sum_{j=1}^{f}\sum_{i=1}^{n}\sum_{k'\neq k}\left(\mathbf{w}_j^{(k')^\top}\mathbf{h}_{k,i}\right)^2 \tag{50}$$

$$+ \frac{\lambda_{\mathbf{W}_0}}{2}\sum_{k=1}^{K}\|\mathbf{w}_k\|_2^2 + \frac{\lambda_H}{2}\sum_{k=1}^{K}\sum_{i=1}^{n}\|\mathbf{h}_{k,i}\|_2^2 \tag{51}$$

$$\overset{(a)}{\geq} \frac{1}{2Kfn}\sum_{k=1}^{K}\sum_{j=1}^{f}n\frac{1}{n}\sum_{i=1}^{n}\left(\mathbf{w}_j^{(k)^\top}\mathbf{h}_{k,i} - 1\right)^2 + \frac{\lambda_{\mathbf{W}_0}}{2}\sum_{k=1}^{K}\|\mathbf{w}_k\|_2^2 + \frac{\lambda_H}{2}\sum_{k=1}^{K}n\frac{1}{n}\sum_{i=1}^{n}\|\mathbf{h}_{k,i}\|_2^2 \tag{52}$$

$$\overset{(b)}{\geq} \frac{1}{2Kfn}\sum_{k=1}^{K}\sum_{j=1}^{f}n\left(\mathbf{w}_j^{(k)^\top}\frac{1}{n}\sum_{i=1}^{n}\mathbf{h}_{k,i} - 1\right)^2 + \frac{\lambda_{\mathbf{W}_0}}{2}\sum_{k=1}^{K}\left\|\mathbf{w}_j^{(k)}\right\|_2^2 + \frac{\lambda_H}{2}\sum_{k=1}^{K}n\left\|\frac{1}{n}\sum_{i=1}^{n}\mathbf{h}_{k,i}\right\|_2^2 \tag{53}$$

The inequality $(a)$ follows from setting

$$\mathbf{w}^{(k')^\top}\mathbf{h}_{k,i} = 0 \tag{54}$$

for all $k' \neq k$ and $i \in [n]$. In $(b)$ we used Jensen's inequality, which (due to the strict convexity of $(\cdot - 1)^2$ and $\|\cdot\|^2$) holds with equality iff

$$\mathbf{h}_{k,1} = \ldots = \mathbf{h}_{k,n} \tag{55}$$

for all $k \in [K]$.

Since all features in each class are identical, $\mathbf{h}_k = \mathbf{h}_{k,i_k}$ for all $i_k \in [n_k]$. Continuing from the last inequality, we have

$$RHS \overset{(c)}{\geq} \frac{1}{2Kfn}\sum_{k=1}^{K}nf\left(\frac{1}{f}\sum_{j=1}^{f}\mathbf{x}_j^{(k)} - 1\right)^2 + \frac{\lambda_{\mathbf{W}_0}}{2}K\left(\frac{1}{K}\sum_{k=1}^{K}\|\mathbf{w}_k\|_2\right)^2 + \frac{n\lambda_H}{2}K\left(\frac{1}{K}\sum_{k=1}^{K}\|\mathbf{h}_k\|_2\right)^2 \tag{56}$$

We get $(c)$ by Jensen's inequality, which holds with equality iff

$$\overline{\mathbf{w}}^{(k)^\top}\mathbf{h}_k = \mathbf{w}_1^{(k)^\top}\mathbf{h}_k = \mathbf{w}_2^{(k)^\top}\mathbf{h}_k = \ldots = \mathbf{w}_f^{(k)^\top}\mathbf{h}_k, \forall j \in [f] \tag{57}$$

$$\|\mathbf{w}_1\|_2 = \ldots = \|\mathbf{w}_K\|_2, \tag{58}$$

$$\|\mathbf{h}_1\|_2 = \ldots = \|\mathbf{h}_K\|_2, \tag{59}$$

then continuing from the RHS of the last inequality

$$RHS \overset{(d)}{\geq} \frac{1}{2}\left(\frac{1}{K}\sum_{k=1}^{K}\mathbf{x}^{(k)} - 1\right)^2 + K\sqrt{n\lambda_H\lambda_{\mathbf{W}_0}}\|\mathbf{w}_k\|_2\|\mathbf{h}_k\|_2 \tag{60}$$

In $(d)$ we use Jensen inequality for the first term, the equality holds when

$$\overline{\mathbf{w}}^{(1)^\top}\mathbf{h}_1 = \overline{\mathbf{w}}^{(2)^\top}\mathbf{h}_2 \ldots = \overline{\mathbf{w}}^{(K)^\top}\mathbf{h}_K; \tag{61}$$

and young's inequality $\frac{a}{2} + \frac{b}{2} \geq \sqrt{ab}$ for second and third term, with $a = \lambda_{\mathbf{W}_0} \left( \frac{1}{K} \sum_{k=1}^{K} \|\mathbf{w}_k\|_2 \right)^2$ and $b = n\lambda_H \left( \frac{1}{K} \sum_{k=1}^{K} \|\mathbf{h}_k\|_2 \right)^2$. It holds with equality iff

$$\lambda_{\mathbf{W}_0} \|\mathbf{w}_k\|_2^2 = n\lambda_H \|\mathbf{h}_k\|_2^2. \tag{62}$$

Note the sequel of equality conditions are satisfied by null solution $(\mathbf{V}, \mathbf{W}_0^*, \mathbf{H}^*) = (\mathbf{0}, \mathbf{0}, \mathbf{0})$, so it remains to show when the solution is not trivial.

According to the equality conditions and the symmetry w.r.t. $k \in [k]$ and $j \in [f]$, the RHS of the last inequality turns into the expression

$$\frac{1}{2} \left( \mathbf{w}_j^{(k)} \mathbf{h}_k - 1 \right)^2 + K\sqrt{n\lambda_H \lambda_{\mathbf{W}_0}} \|\mathbf{w}_k\|_2 \|\mathbf{h}_k\|_2 \tag{63}$$

$$= \frac{1}{2} \left( \left\| \mathbf{w}_j^{(k)} \right\|_2 \|\mathbf{h}_k\|_2 \cos\alpha - 1 \right)^2 + K\sqrt{n\lambda_H \lambda_{\mathbf{W}_0}} \|\mathbf{w}_k\|_2 \|\mathbf{h}_k\|_2, \tag{64}$$

$$= \frac{1}{2} \left( \sqrt{\frac{\lambda_{\mathbf{W}_0}}{n\lambda_H}} \|\mathbf{w}_k\|_2^2 \cos\alpha - 1 \right)^2 + K\lambda_{\mathbf{W}_0} \|\mathbf{w}_k\|_2^2 \tag{65}$$

$$= \frac{1}{2} \frac{\lambda_{\mathbf{W}_0}}{n\lambda_H} \cos^2\alpha \left( \|\mathbf{w}_k\|_2^2 + \frac{K\lambda_{\mathbf{W}_0} - \sqrt{\frac{\lambda_{\mathbf{W}_0}}{n\lambda_H}} \cos\alpha}{\frac{\lambda_{\mathbf{W}_0}}{n\lambda_H} \cos^2\alpha} \right)^2 + \frac{1}{2} - \frac{1}{2} \frac{\left( K\lambda_{\mathbf{W}_0} - \sqrt{\frac{\lambda_{\mathbf{W}_0}}{n\lambda_H}} \cos\alpha \right)^2}{\frac{\lambda_{\mathbf{W}_0}}{n\lambda_H} \cos^2\alpha} \tag{66}$$

which obtains minimum at $\|\mathbf{w}_k\|^2 = 0$ if $K\lambda_{\mathbf{W}_0} - \sqrt{\frac{\lambda_{\mathbf{W}_0}}{n\lambda_H}} \cos\alpha \geq 0$ and $\|\mathbf{w}_k\|^2 = \frac{-K\lambda_{\mathbf{W}_0} + \sqrt{\frac{\lambda_{\mathbf{W}_0}}{n\lambda_H}} \cos\alpha}{\frac{\lambda_{\mathbf{W}_0}}{n\lambda_H} \cos^2\alpha}$ if $K\lambda_{\mathbf{W}_0} - \sqrt{\frac{\lambda_{\mathbf{W}_0}}{n\lambda_H}} \cos\alpha < 0$, where $\alpha = \angle(\mathbf{w}_j^{(k)}, \mathbf{h}_k)$, and for simplicity we denote

$$f(\alpha) := \frac{1}{2} - \frac{1}{2} \frac{\left( K\lambda_{\mathbf{W}_0} - \sqrt{\frac{\lambda_{\mathbf{W}_0}}{n\lambda_H}} \cos\alpha \right)^2}{\frac{\lambda_{\mathbf{W}_0}}{n\lambda_H} \cos^2\alpha}. \tag{67}$$

In the first case, the minimum is $\frac{1}{2}$; in the second case, we have to find the smallest value of $f(\alpha)$, which is equivalent to find the lower bound (or upper bound) of $\alpha$ (or $\cos\alpha$) since $f(\alpha)$ is increasing in $\alpha$ if $K\lambda_{\mathbf{W}_0} - \sqrt{\frac{\lambda_{\mathbf{W}_0}}{n\lambda_H}} \cos\alpha < 0$.

We observe that $\alpha$ is non-zero, or otherwise by the equality conditions from above, $\mathbf{h}_k$ has to align with all $\mathbf{w}_j^{(k)}$ which is impossible.

since $\left\| \mathbf{w}_j^{(k)} \right\|$ are equiangular and equally normed, for any norm-fixed vector $\mathbf{q} \in \mathbb{R}^d$ satisfying $\langle \mathbf{w}_1^{(k)}, \mathbf{v} \rangle = \ldots = \langle \mathbf{w}_f^{(k)}, \mathbf{v} \rangle$, we have

$$\min \alpha \Leftrightarrow \max_{\alpha} \cos\alpha \Leftrightarrow \min_{\alpha} \sum_{j=1}^{f} \left\| \mathbf{w}_j^{(k)} - \mathbf{q} \right\|^2 \tag{68}$$

which has its optimum when $\mathbf{q}$ aligns with $\overline{\mathbf{w}}^{(k)}$ according to the convexity of the last minimization problem, and thus $\cos\alpha^* = \sqrt{\frac{\cos^2\theta + f\sin^2\theta}{f}}$, the optimal direction also form an angle $\rho = \arccos \frac{f\sin\theta}{\sqrt{\cos^2\theta + f\sin^2\theta}}$ with $\mathbf{w}_k$.

We have bound $\alpha \in [K\sqrt{n\lambda_H\lambda_{\mathbf{W}_0}}, \sqrt{\frac{\cos^2\theta + f\sin^2\theta}{f}}]$. the minimum of the objective is achieved at the boundaries. It is easy to see that the value of the objective function is lower than $\frac{1}{2}$, which is the value at the right boundary. Therefore, at the optimum, $\cos\alpha^* = \sqrt{\frac{\cos^2\theta + f\sin^2\theta}{f}}$ and $\cos\rho = \frac{f\sin\theta}{\sqrt{\cos^2\theta + f\sin^2\theta}}$

For the orthogonality of $\mathbf{h}_k$'s and $\mathbf{w}_k$'s, since let $\mathbf{h}_k = \gamma_1\overline{\mathbf{w}}^{(k)}$ and $\mathbf{h}_{k'} = \gamma_2\overline{\mathbf{w}}^{(k')}$, since $\mathbf{w}^{(k')\top}\mathbf{h}_k = 0$, $\langle\mathbf{h}_k, \mathbf{h}_{k'}\rangle = \gamma_2\frac{1}{f}\sum_{j=1}^{f}\langle\mathbf{w}^{(k')\top}, \mathbf{h}_k\rangle = 0$. Then we use definition of $\mathbf{w}_j^{(k)}$ to extend the $0 = \langle\mathbf{h}_k, \mathbf{h}_{k'}\rangle = \frac{1}{f^2}\langle\mathbf{w}_k, \mathbf{w}_{k'}\rangle\sin^2\theta$ which implies $\langle\mathbf{w}_k, \mathbf{w}_{k'}\rangle = 0$. We also have learnt that

$$\|\mathbf{h}_k\|_2^2 = \frac{\lambda_{\mathbf{W}_0}}{n\lambda_H}\|\mathbf{w}_k\|_2^2 = \frac{-K\lambda_{\mathbf{W}_0} + \sqrt{\frac{\lambda_{\mathbf{W}_0}}{n\lambda_H}}\cos\alpha^*}{\cos^2\alpha^*}. \tag{69}$$

$\square$

The imbalanced case is proved with a slightly different strategy.

*Proof of Theorem 3.4.* In the imbalanced case

$$\frac{1}{2S}\|\mathbf{WH} - \mathbf{Y}\|_F^2 + \frac{\lambda_H}{2}\|\mathbf{H}\|_F^2 + \frac{\lambda_{\mathbf{W}_0}}{2}\|\mathbf{W_0}\|_F^2 \tag{70}$$

$$\overset{(a)}{\geq} \frac{1}{2S}\sum_{k=1}^{K}\sum_{i=j}^{f_k}n_k\frac{1}{n_k}\sum_{i=1}^{n_k}\left(\mathbf{w}_j^{(k)\top}\mathbf{h}_{k,i} - 1\right)^2 + \frac{\lambda_H}{2}\sum_{k=1}^{K}n_k\frac{1}{n_k}\sum_{i=1}^{n_k}\|\mathbf{h}_{k,i}\|_2^2 + \frac{\lambda_{\mathbf{W}_0}}{2}\sum_{k=1}^{K}\|\mathbf{w}_k\|_2^2 \tag{71}$$

$$\overset{(b)}{\geq} \frac{1}{2S}\sum_{k=1}^{K}\sum_{j=1}^{f_k}n_k\left(\mathbf{w}_j^{(k)\top}\frac{1}{n_k}\sum_{i=1}^{n_k}\mathbf{h}_{k,i} - 1\right)^2 + \frac{\lambda_H}{2}\sum_{k=1}^{K}n_k\left\|\frac{1}{n_k}\sum_{i=1}^{n_k}\mathbf{h}_{k,i}\right\|_2^2 + \frac{\lambda_{\mathbf{W}_0}}{2}\sum_{k=1}^{K}\|\mathbf{w}_k\|_2^2 \tag{72}$$

$$\overset{(c)}{\geq} \frac{1}{2S}\sum_{k=1}^{K}n_kf_k\left(\overline{\mathbf{w}}^{(k)\top}\mathbf{h}_k - 1\right)^2 + \frac{\lambda_H}{2}\sum_{k=1}^{K}n_k\|\mathbf{h}_k\|_2^2 + \frac{\lambda_{\mathbf{W}_0}}{2}\sum_{k=1}^{K}\|\mathbf{w}_k\|_2^2 \tag{73}$$

$$\geq \frac{1}{2S}\min_{\mathbf{H},\mathbf{W}}\sum_{k=1}^{K}n_k\left[f_k\left(\overline{\mathbf{w}}^{(k)\top}\mathbf{h}_k - 1\right)^2 + \lambda_HS\|\mathbf{h}_k\|_2^2 + \frac{\lambda_{\mathbf{W}_0}S}{n_k}\|\mathbf{w}_k\|_2^2\right] \tag{74}$$

$$\overset{(d)}{=} \frac{1}{2S}\sum_{k=1}^{K}n_k\min_{\mathbf{h}_k,\mathbf{w}_k}\left[f_k\left(\overline{\mathbf{w}}^{(k)\top}\mathbf{h}_k - 1\right)^2 + \lambda_HS\|\mathbf{h}_k\|_2^2 + \frac{\lambda_{\mathbf{W}_0}S}{n_k}\|\mathbf{w}_k\|_2^2\right] \tag{75}$$

$$\overset{(e)}{\geq} \frac{1}{2S}\sum_{k=1}^{K}n_k\min_{\mathbf{h}_k,\mathbf{w}_k}\left[f_k\left(\overline{\mathbf{w}}^{(k)\top}\mathbf{h}_k - 1\right)^2 + 2S\sqrt{\frac{\lambda_H\lambda_{\mathbf{W}_0}}{n_k}}\|\mathbf{h}_k\|\|\mathbf{w}_k\|\right] \tag{76}$$

where $(a)$, $(b)$, and $(c)$ are the same as the last proof. The equality of $(e)$ holds only when
$$\lambda_{\mathbf{W}_0}\|\mathbf{w}_k\|^2 = n_k\lambda_{\mathbf{H}}\|\mathbf{h}_k\|^2$$
by Young's Inequality. We decompose the objective in $(e)$ different from that in the proof of theorem 3.3 because the existence of non-identical $f_k$ fails optimal condition (61). Now we minimize

$$\left[f_k\left(\overline{\mathbf{w}}^{(k)\top}\mathbf{h}_k - 1\right)^2 + 2S\sqrt{\frac{\lambda_H\lambda_{\mathbf{W}_0}}{n_k}}\|\mathbf{h}_k\|\|\mathbf{w}_k\|\right]$$

for each $k$, where the method in the last proof is applicable, which result in

$$\|\mathbf{w}_k\|^2 = \frac{-\frac{S\lambda_{W_0}}{f_kn_k} + \sqrt{\frac{\lambda_{W_0}}{n_k\lambda_H}}\cos\alpha_k^*}{\frac{\lambda_{W_0}}{n_k\lambda_H}\cos^2\alpha_k^*} \tag{77}$$

when $\cos\alpha_k^* > \frac{S}{f_k}\sqrt{\frac{\lambda_{W_0}\lambda_H}{n_k}}$

$\square$

The reader can check that the result of the imbalanced setting contains the balanced one as a special case.

When $W$ and $W_0$ are fixed, the optimal conditions become simpler.

*Proof of Corollary 3.7.* Similar to the proof of Theorem 3.4,

$$\frac{1}{2S}\|\mathbf{W}\mathbf{H} - \mathbf{Y}\|_F^2 + \frac{\lambda_H}{2}\|\mathbf{H}\|_F^2 \tag{78}$$

$$\overset{(a)}{\geq} \frac{1}{2S}\sum_{k=1}^{K}\sum_{i=j}^{f_k} n_k \frac{1}{n_k}\sum_{i=1}^{n_k}\left(\mathbf{w}_j^{(k)\top}\mathbf{h}_{k,i} - 1\right)^2 + \frac{\lambda_H}{2}\sum_{k=1}^{K} n_k \frac{1}{n_k}\sum_{i=1}^{n_k}\|\mathbf{h}_{k,i}\|_2^2 \tag{79}$$

$$\overset{(b)}{\geq} \frac{1}{2S}\sum_{k=1}^{K}\sum_{j=1}^{f_k} n_k \left(\mathbf{w}_j^{(k)\top}\frac{1}{n_k}\sum_{i=1}^{n_k}\mathbf{h}_{k,i} - 1\right)^2 + \frac{\lambda_H}{2}\sum_{k=1}^{K} n_k \left\|\frac{1}{n_k}\sum_{i=1}^{n_k}\mathbf{h}_{k,i}\right\|_2^2 \tag{80}$$

$$\overset{(c)}{\geq} \frac{1}{2S}\sum_{k=1}^{K} n_k f_k\left(\overline{\mathbf{w}}^{(k)\top}\mathbf{h}_k - 1\right)^2 + \frac{\lambda_H}{2}\sum_{k=1}^{K} n_k\|\mathbf{h}_k\|_2^2 \tag{81}$$

$$\overset{(d)}{\geq} \frac{1}{2S}\sum_{k=1}^{K} n_k \min_{\mathbf{h}_k}\left[f_k\left(\overline{\mathbf{w}}^{(k)\top}\mathbf{h}_k - 1\right)^2 + \lambda_H S\|\mathbf{h}_k\|_2^2\right], \tag{82}$$

where $\overline{\mathbf{w}}^{(k)\top}\mathbf{h}_k = \mathbf{w}_j^{(k)\top}\mathbf{h}_k = \|\mathbf{h}_k\|_2\cos\alpha_k$ for all $j\in[f_k]$ and $\alpha_k$ is the angle between $\mathbf{h}_k$ and $\mathbf{w}_j^{(k)}$ for every $j\in[f_k]$. from (a) to (c), the equality holds iff

$$\mathbf{h}_k^* = \mathbf{h}_{k,1}^* = \ldots = \mathbf{h}_{k,n_k}^*, \forall k\in[K] \tag{83}$$

$$\langle\mathbf{w}_{k'}^*, \mathbf{h}_k^*\rangle = 0, \forall k'\neq k\in[K] \tag{84}$$

$$\langle\mathbf{h}_{k'}^*, \mathbf{h}_k^*\rangle = 0, \forall k'\neq k \tag{85}$$

$$\mathbf{w}_1^{(k)\top}\mathbf{h}_k^* = \ldots = \mathbf{w}_{f_k}^{(k)\top}\mathbf{h}_k^*, \forall K\in[K] \tag{86}$$

Since each summand is the minimum of a convex function of $\|\mathbf{h}_k\|_2$, $\mathbf{P}$ attains its minimum when all of the summands are minimized separately in (d), that is, by the analogous analysis to Theorem 3.3, $\|\mathbf{h}_k^*\|_2 = \frac{f_k\cos\alpha_k^*}{f_k\cos^2\alpha_k^* + \lambda_H S}$. $\qquad\square$

**Since we only use the information of $\theta$ at the end of the proof, this generalized NC can hold also true given $\theta_k$'s have different values for each $k\in[K]$** in the construction of Multi-Center Frame, demonstrating the generality of our method. However, in this paper we are limited to the case of identical $\theta$ for all classes.

# E  Derivative of the Cosine Loss

In this section we calculate the derivative of the cosine loss $\mathrm{Cos}(\mathbf{w}, \mathbf{h})$. For $\mathrm{Cos}(\mathbf{w}, \mathbf{h}) = \|\langle\mathbf{w}, \frac{\mathbf{h}}{\|\mathbf{h}\|}\rangle - 1\|^2$,

$$\frac{\mathrm{d}\,\mathrm{Cos}(\mathbf{w}, \mathbf{h})}{\mathrm{d}\mathbf{h}} = -2\cdot(1-a)\mathbf{J}\mathbf{w},$$

where $\mathbf{J}$ is the Jacobian of $\frac{\mathbf{h}}{\|\mathbf{h}\|}$ w.r.t. $\mathbf{h}$ and $a = \frac{\langle\mathbf{w},\mathbf{h}\rangle}{\|\mathbf{h}\|}$ It is straightforward to calculate $\mathbf{J} = \frac{1}{\|\mathbf{h}\|^3}(\|\mathbf{h}\|\,\mathbf{I} - \mathbf{h}\mathbf{h}^\top)$. $\|\mathbf{h}\|\,\mathbf{I} - \mathbf{h}\mathbf{h}^\top$ has eigenvalue $\|\mathbf{h}\| - \|\mathbf{h}\|^2$ on the direction $\frac{\mathbf{h}}{\|\mathbf{h}\|}$ and $\|\mathbf{h}\|$ on all

other directions orthogonal to $\frac{\mathbf{h}}{\|\mathbf{h}\|}$, so that

$$\mathbf{Jw} \tag{87}$$

$$=\mathbf{J}(\mathbf{w} - a\frac{\mathbf{h}}{\|\mathbf{h}\|} + a\frac{\mathbf{h}}{\|\mathbf{h}\|}) \tag{88}$$

$$=\frac{1}{\|\mathbf{h}\|^2}(\mathbf{w} - a\frac{\mathbf{h}}{\|\mathbf{h}\|}) + a(\frac{1}{\|\mathbf{h}\|^2} - \frac{1}{\|\mathbf{h}\|})\frac{\mathbf{h}}{\|\mathbf{h}\|} \tag{89}$$

$$=\frac{1}{\|\mathbf{h}\|^2}(\mathbf{w} - a\mathbf{h}). \tag{90}$$

# F   Implementation Details and Results

## F.1   Long-Tail Classification on Four Datasets

We run experiments with backbone ResNet50 and DenseNet150 on the four datasets (CIFAR-10, CIFAR-100, SVHN and STL-10) by 1 A100 GPU, and run ResNet50 on ImageNet by 2 A100 GPU with an extra linear layer that expand the dimension of the backbone feature to be larger than $fK$. We use DenseNet150 with a reduction 0.5; the growth rate is set 12 on CIFAR-10 and CIFAR-100 and 32 on SVHN and STL-10. The reason we adjust growth rate to make the dimension of output feature higher than $fK$) which is necessary for our method. To display the neural collapse phenomenon, an extra linear layer is added between the backbone and the classifier.

We use the code released by [73] to produce the imbalanced datasets. We train the model on the four dataset for 200 epochs, with a step learning rate initialized to 0.1 decaying to 0.01 and 0.001 at epoch 160 and epoch 180, batch size of 128, a momentum of 0.9, and a weight decay of $2e - 4$. We train on ImageNet for 200 epochs, with a CosineAnnealing learning rate initialized to 0.1, batch size of 512, a momentum of 0.9, and a weight decay of $5e - 4$.

Given imbalance ratio $\tau = \frac{n_{\min}}{n_{\max}}$, where $n_{\min}$ and $n_{\max}$ are the minimal and maximal numbers of training samples in all classes, the numbers of training samples are decayed exponentially from $n_{\max}$ to $n_{\max}$ among classes. We take the canonical data normalization and augmentation for the five datasets. We use both the regular method and Mixup method [82] to demonstrate the effectiveness of our proposed structure. The hyper-parameter $a = 1$ by default controls the shape of the symmetric $\beta$ distribution when Mixup is used. We fix the classifiers as orthonormal vectors for every setting.

We train the network by weighted cosine loss with a norm regularization which has the form

$$\mathcal{L}(\mathbf{h}_{k,i_k}, \mathbf{W}) = \frac{1}{B}\sqrt{\frac{\cos^2\theta + f_k\sin^2\theta}{f_k}}\left(\sum_{j=1}^{f_k}\mathrm{Cos}(\mathbf{w}_j^{(k)}, \mathbf{h}_{k,i_k})\right) + \lambda\left(\|\mathbf{h}_{k,i_k}\| - 1\right)^2, \tag{91}$$

where $\mathbf{h}_{k,i_k}$ is an example in class $k$, and $\lambda = 6e - 4$.

The classification rule of our method, as suggested by remark 3.9 is $c = argmax_{k\in[K]}\left\langle\frac{BN(\mathbf{h})}{\|BN(\mathbf{h})\|}, \mathbf{w}_k\right\rangle$ where $BN(\mathbf{h})$ is the batch normalization of $\mathbf{h}$; we use $BN$ since the data may have non-zero mean; For SVHN and STL-10 datasets, we first normalize the backbone outputs $\mathbf{h}$, then train and predict by the same loss and decision rule. For Densenet, we use the $BN(\frac{\mathbf{h}}{\|\mathbf{h}\|})$ as the backbone output, apply the original loss to train, and predict by the same decision rule for ResNet50. The code is modified from [26], and can be found in the supplementary

material. Then the upper bound of the gradient norm of a class $k$ becomes

$$\left\| \sum_{i_k}^{n_k} \sqrt{\frac{\cos^2 \theta + f_k \sin^2 \theta}{f_k}} \sum_j^{f_k} \frac{d \operatorname{Cos}(\mathbf{w}_j^{(k)}, \mathbf{h}_{k,i_k})}{d\mathbf{h}_{k,i_k}} \right\|_2 \tag{92}$$

$$\leq \sum_{k_i}^{n_k} \|\mathbf{h}_{k,i_k}\|^{-2} \sqrt{\frac{\cos^2 \theta + f_k \sin^2 \theta}{f_k}} \left\| \sum_j^{f_k} \mathbf{w}_j^{(k)} \right\|_2 \tag{93}$$

$$\leq \sum_{k_i}^{n_k} \|\mathbf{h}_{k,i_k}\|^{-2} \left(\cos^2 \theta + f_k \sin^2 \theta\right). \tag{94}$$

Compared with the upper bound of the gradient norm of the unweighted cosine loss

$$\left\| \sum_{i_k}^{n_k} \sum_j^{f_k} \frac{d \operatorname{Cos}(\mathbf{w}_j^{(k)}, \mathbf{h}_{k,i_k})}{d\mathbf{h}_{k,i_k}} \right\|_2 \tag{95}$$

$$\leq \sum_{k_i}^{n_k} \|\mathbf{h}_{k,i_k}\|^{-2} \left\| \sum_j^{f_k} \mathbf{w}_j^{(k)} \right\|_2 \tag{96}$$

$$\leq \sum_{k_i}^{n_k} \|\mathbf{h}_{k,i_k}\|^{-2} \sqrt{f_k \cos^2 \theta + f_k^2 \sin^2 \theta}. \tag{97}$$

Our weighted loss gives the linearity between the upper bound $f_k$ and the gradient norm.

Figure 4 presents the NC phenomenon under objective $\mathbf{P}$ with or without regularization on the feature norm.

Figure 5 shows NCMC for different architectures, including DenseNet, ResNet, VGG, and LeNet.

Table 4: Loss $\mathbf{P}$ vs Cosine Regression Loss on CIFAR-100

| $\tau$ | 0.005 | 0.01 | 0.02 |
|---|---|---|---|
| $\mathbf{P}$ w/o mixup | 41.9± 0.2 | 43.4 ± 0.3 | 43.5±0.2 |
| $\mathbf{P}$ w mixup | 36.5±0.6 | 40.1±0.4 | 49.0±0.1 |
| CAL | **46.3**±0.3 | **50.1**±0.2 | **54.0**±0.2 |

**Table** 5 presents long-tail classification results on SVHN and STL-10, complementary to **Table** 2.

**Table** 6 shows how the accuracy of imbalanced learning is changed by the parameters $f$, and $\theta$ on CIFAR-100.

### F.2  Selection of Hyperparameters

When picking $f_k$ according to the class-aware strategy, we hope $f_m n_m \approx f_l n_l$ for $l, m \in [K]$. This condition, combined with the equality $fK = \sum_{k=1}^k f_k n_k$ refuses small $f$ when there is a heavy class imbalance. For example, when the $K$ classes are balanced, the strategy requires $f_m \approx f_l$ for $l, m \in [K]$, and thus $f_k \approx f$ by the equality; on the other hand, for extreme imbalance, say if the ratio $\tau \leq \frac{1}{bK}$ with $b$ a large constant, we have $fK > f_K \geq bK$, thus $f > b$. As $f$ increases from 10 to 20, imbalanced ratio $\tau = 0.005$ (the heaviest ratio in the imbalanced settings), and $\theta = 0.2$, we achieve $80.6 \pm 0.5$ on cifar10, and $46.6 \pm 0.4$, both better than the reported in **Table** 2.

Moreover, we suggest small $\theta \in (0, \pi/4)$ in practice (e.g., $\cos \theta \geq \frac{\sqrt{2}}{2}$), since large $\theta$ generates concentrated centers of the classes, which is more likely to overweight and overfit the minor classes (consider the case of $\cos \theta = 1$ where all the centers collapse to one direction, then the gradient directions will be severely skewed).

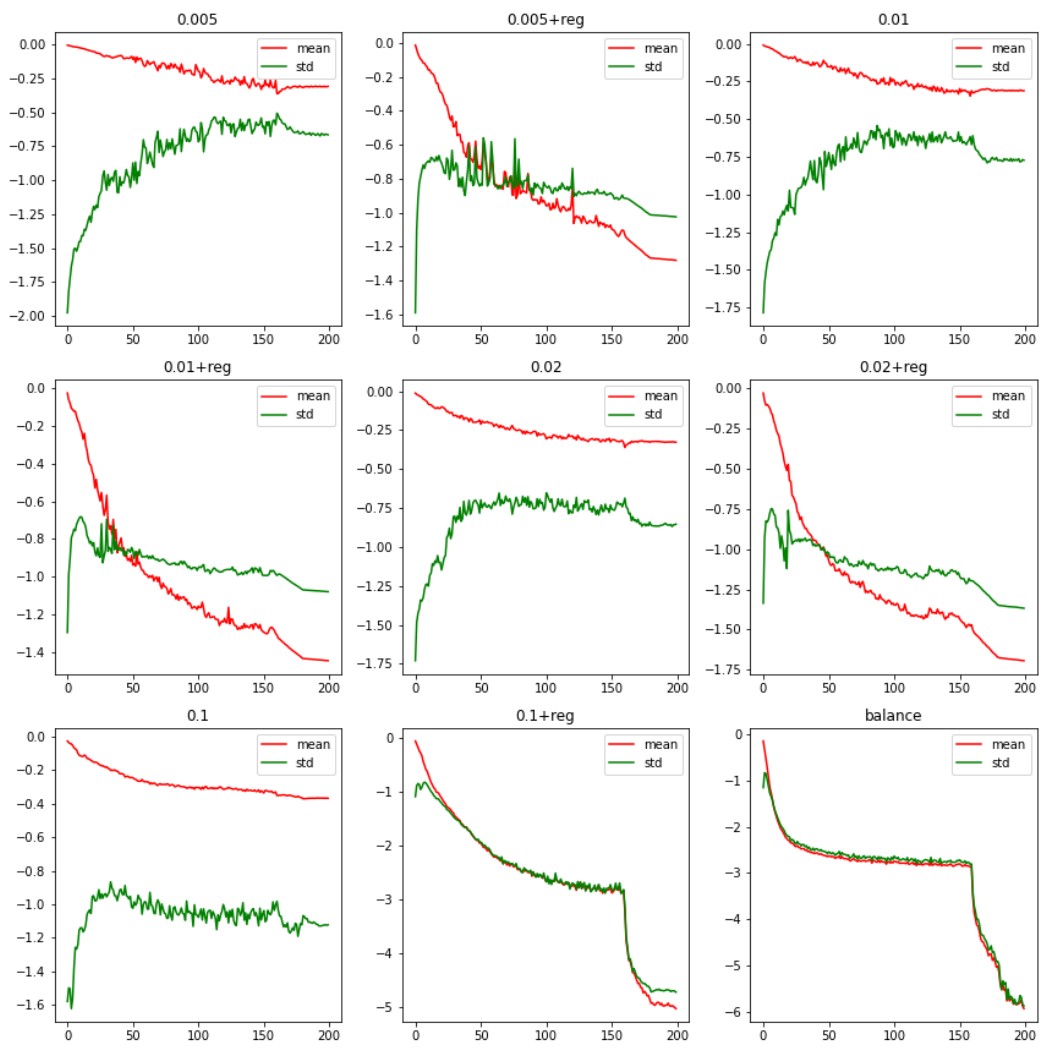

Figure 4: NCMC phenomenon under Loss $\mathbf{P}$ at different epochs. We draw the mean and standard deviation of the neural collapse metric used in the paper on CIFAR-10 for different imbalanced ratios w/ or w/o regularization. The horizontal axis is $\log_{10}$-scaled NC metric value. The regularization coefficient is 5e-4.

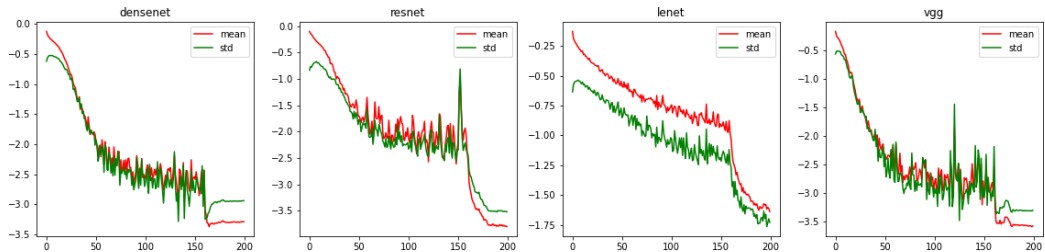

Figure 5: NCMC phenomenon in Different Backbones at different epochs. We draw the mean and standard deviation of the neural collapse metric used in the paper for four different backbones that trained on Cifar10-LT with $\tau = 0.005$: DenseNet150, ResNet50, LeNet, and, VGG11. The horizontal axis is $\log_{10}$-scaled NC metric value.

Table 5: Long-tailed classification accuracy (%) with ResNet and DenseNet on SVHN and STL-10.

| Methods | SVHN | | | STL-10 | | |
|---|---|---|---|---|---|---|
| | 0.005 | 0.01 | 0.02 | 0.005 | 0.01 | 0.02 |
| *ResNet* | | | | | | |
| CE | 39.4±0.2 | 40.6±0.2 | 46.4±0.2 | 33.6 ±0.1 | 35.0±0.3 | 36.3±0.3 |
| SETF | 41.3±0.2 | 45.4±0.1 | **49.6**±0.2 | 37.4±0.4 | 38.2±0.7 | 42.0 ±0.3 |
| CAL | **43.5**±0.1 | **47.4**±0.2 | 49.1±0.1 | **40.5**±0.2 | **42.8**±0.4 | **45.4**±0.3 |
| *DenseNet* | | | | | | |
| CE | 38.9±0.4 | 40.8±0.4 | 47.2±0.3 | 38.5±0.6 | 41.2 ±0.3 | 44.9±0.2 |
| SETF | 40.5±0.1 | 44.8±0.2 | 48.4±0.2 | 39.5±0.3 | 42.9±0.3 | 46.3±0.2 |
| CAL | **45.4**±0.1 | **46.6**±0.1 | **48.8**±0.2 | **42**±0.3 | **43.3**±0.4 | **47.4**±0.1 |

Table 6: ResNet50's accuracy changes with the parameter tuple $(f,\theta)$ on CIFAR100, $\tau = 0.005$

| $(f,\theta)$ | 0 | 0.2 | 0.4 | 0.6 | 0.8 | 1 | $\pi/2$ |
|---|---|---|---|---|---|---|---|
| 4 | 1.8 | 41.8 | 42.2 | 43.1 | 41.8 | 41.6 | 44.0 |
| 7 | 1.1 | 45.2 | 45 | 44.4 | 41.6 | 43.2 | 43.7 |
| 10 | 1.3 | 45.7 | 46.1 | 44.4 | 43.5 | 42.6 | 43.0 |
| 13 | 2.3 | 46.0 | 46.2 | 44.5 | 42.7 | 41.8 | 41.5 |
| 16 | 0.9 | 46 | 44.9 | 42.7 | 41.1 | 39.6 | 38.6 |
| 20 | 1.6 | **47.0** | 44.7 | 41.3 | 39.1 | 38.2 | 37.9 |

# G   Heatmaps of the Neural Collapse

Figure 6 shows the inner product of normalized class-mean features during training. We observe that (a) records the contracted mean features at initialization; then the training separates the class means gradually to be orthogonal.

We also capture the feature collapse of the most minor class (class size=25) for CIFAR-10 dataset in Fig 7. It is interesting to note that the collapse occurs almost at the beginning.

# H   Impact and Limitations

**Impact of our work.** 1. Our topic is Imbalanced Classification which is a general concern in machine learing.

2. We consider a classification rule that work better than regular classification rule under certain theoretical assumptions in the imbalanced setting. The analysis of "hard-to-predict feature" is novel and can be further developed in general machine learning theory.

3. We have studied imbalanced learning through the lens of Neural Collapse, which provides insight into the connection among the optimal feature-classifier alignment, the classification rule, and the performance of DNN.

4. We proposed a loss and the strategy for fixed classifier that has comparable performance to methods with learnable classifier; the loss and strategy can apply to general machine learning models.

**Limitations.**

1. The theoretical analysis of our motivation only considers Gaussian case;

2. Due to the computationally expensive optimization on the Stiefel manifold for large dimension $fK$, the UFM analysis on learnable classifier in Theorem 3.3 and 3.4 are not justified in practical networks;

3. Although the proposed loss is applicable to general classification models, we only conduct experiments on two architectures, ResNet and DenseNet;

4. The proposed method for imbalanced learning does not have impressive performance on high-dimensional datasets (large number of classes, large number of features), one of the possible reasons is the use of global expansion factor to expand the backbone feature dimension imposes negative effect on the representation of an deep architecture;

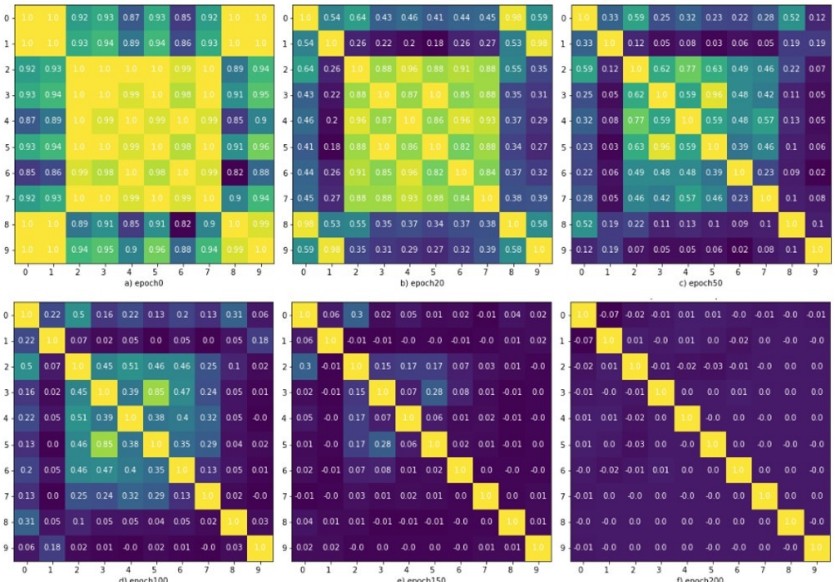

Figure 6: The heatmap of Mean-feature Separateness.

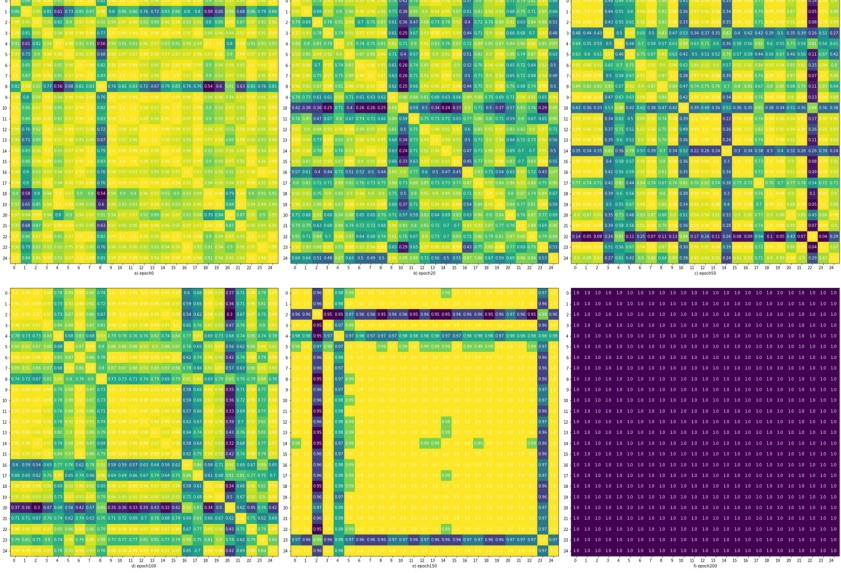

Figure 7: The Within-Class Feature Collapse.

5. Although the class-aware strategy is a novel idea, the implementation is far from optimal. The example given in section 3.5 shows $f_1 n_1 = 4$, $f_2 n_2 = 12$, and $f_3 n_3 = 3$ which will result in an emphasis on the middle class instead of the minor class. How to create a strategy that is more "class-aware" will be our future work.

6. We compare our work with RBL [42] in **Table** 7. It is remarkable that the post-hoc logit adjustment significantly improves RBL. Our method outperforms RBL in two settings for cifar100. We conjecture that in a setting of a small imbalanced ratio and a large number of classes, the hard-to-predict distribution may dominate the performance. The reason that our method CAL has lower accuracy compared to RBL in other settings, is three-fold: 1. Imbalanced learning with MSE loss is less effective than CE loss in general; 2. Our experiment setting is chosen as close as possible to that where the theoretical analysis (proposition B.1 and C.1) is conducted, for example, to match the isotropy of the Gaussian, we normalize/batch normalize the feature to ensure it is unit norm and centered before the classification, and the weight are unit vectors through training. This setting

possibly harms the learnability and flexibility of our model. 3. Under the class-aware MSE loss, we use the original classifier to be the surrogate classification rule of our general classification rule. However, the rule is designed especially for "hard-to-predict" unseen data and thus is not necessarily optimal for the classification of other unseen data. When the hard-to-predict unseen data takes a very small portion of the population, our design may lose its effectiveness. The success of RBL and PLA inspires us to find an optimal classifier for the loss $\mathbf{P}$ and CAL.

Table 7: CAL vs RBL in long-tail classification trained on ResNet50. $f = 20$ and $\theta = 0.2$ are fixed. The values without $\pm$ are that we did not reproduce. "$-$" represents a missing value.

| Methods | Cifar-10 | | | | Cifar-100 | | | |
|---|---|---|---|---|---|---|---|---|
| | 0.005 | 0.01 | 0.02 | 0.1 | 0.005 | 0.01 | 0.02 | 0.1 |
| **RBL** w/o PLA | 73.6 | 78.5±0.3 | 84.3 | 90.7 | – | – | – | – |
| **RBL** | **81.8**±0.5 | **84.9**±0.3 | **87.6**±0.2 | **92.5**±0.3 | 41.7±0.4 | **51.7**±0.2 | 52.4±0.2 | **68.4**±0.1 |
| **CAL** | 80.0±0.5 | 84.1±0.3 | 85.9 ±0.2 | 92.0±0.3 | **46.5**±0.5 | 50.1±0.3 | **54.3**±0.4 | 65.9±0.3 |

