# OpenReview forum: "Neural Collapse To Multiple Centers For Imbalanced Data"
_NeurIPS.cc/2024/Conference — NeurIPS 2024 poster_

### Official Review · Reviewer_sPeJ · 2024-06-26

**Soundness:** 3
**Presentation:** 3
**Contribution:** 4
**Rating:** 6
**Confidence:** 4

**Summary:**

This paper explores Neural Collapse (NC) in the context of imbalanced data, proposing the concept of Neural Collapse to Multiple Centers (NCMC). It establishes that aligning features from minor classes with more directions improves classification accuracy, introducing the Generalized Classification Rule (GCR). The authors design an MSE-type objective and a practical loss function that induces NCMC, achieving performance comparable to classical imbalanced learning methods. Their findings are validated both theoretically and experimentally, offering new insights into the application of NC in imbalanced learning scenarios.

**Strengths:**

+ The paper is clearly written and easy to follow. Complex concepts are systematically explained, and the logical flow is maintained throughout the document, making it accessible to readers with varying levels of expertise in the field.
+ The findings have significant implications for imbalanced learning, offering a new approach that achieves performance comparable to classical methods, thereby advancing the field of deep learning and classification.

**Weaknesses:**

+ Figure 1 should be larger for better visualization for readers.
+ Theorems 3.3 and 3.4 prove that the optimization problem $\mathbf{P}$ can induce the "Neural Collapse to Multiple Centers" solution, but the loss function in $\mathbf{P}$ does not appear in the experiments. It might be better to include the loss function $\mathbf{P}$ in the comparison experiments and compare, analyze, and explain the results.
+ If multiple centers are better than a single center for each class in hard-to-predict feature distribution, why don't the authors directly use the loss function in $\mathbf{P}$ instead of designing a Cosine Regression Loss? The authors should explain this intent.
+ [1], as a study about long-tailed learning and NC, should be compared within this study.
+ I personally think that Proposition 3.1 deserves more space and discussion in the main text, as it serves as a key motivational component for the paper. Feel free to adopt this suggestion; this point does not affect my rating.

[1] Feature Directions Matter: Long-Tailed Learning via Rotated Balanced Representation. Gao Peifeng, Qianqian Xu, Peisong Wen, Zhiyong Yang, Huiyang Shao, Qingming Huang Proceedings of the 40th International Conference on Machine Learning, PMLR 202:27542-27563, 2023.

**Questions:**

see Weaknesses.

**Limitations:**

see Weaknesses.

---

> ### Author Rebuttal · Authors · 2024-08-07
>
> We appreciate the reviewer's recognition of our contribution to the field of deep learning and classification. To convey the meaning of our work more fluently, we answer the questions in the following order.
>
> $\textbf{Response to weaknesses 1 and 5 }:$
>
> We will enlarge the Fig.1 for a better illustration.
>
> We would like to move Proposition B.1 to the Main Result section (section 3) and introduce the notion of $\tilde{w}_j^{(k)}$ more carefully.
>
> $\textbf{Response to weakness 3}:$
>
> We are sorry that our narrative creates a misunderstanding here.
> In the analysis of the classification rule we are assuming the classifiers and features are well-trained, the optimal symmetric final structure can be seen as the guarantee of the isotropy of the hard-to-predict populations. However, when we use GCR to supervise the model, the training will quickly encounter a gradient vanishing. Indeed, at initialization let $s = argmax_j\lbrace \langle w_j^{(k)},h_{k,i}\rangle\rbrace$, the gradient descent will take $h_{k,i}$ to the direction closer to the specific center $w_s^{(k)}$. Then during training $h_{k,i}$ approaches $w_s^{(k)}$ even faster since $s = argmax_j\lbrace \langle w_j^{(k)},h_{k,i}\rangle\rbrace$ from the beginning. So the model ends up with a structure far from optimal; each class will have several clusters scattered randomly nearly orthogonal to each other. In this case, the model overfits the rule easily. This is the drawback of directly using GCR as a practical objective and the reason we try to use the average of centers instead of the maximum of them.
>
> Both P and Cosine loss are approximations to the Generalized Classification Rule (GCR) while cosine loss is a concise version of P; Both of them use multiple centers, please recall that a center is a vector associated with a certain classifier. The difference between P and Cosine loss is: P uses all centers available while Cosine loss uses centers of one class (the class to which the feature belongs). Unlike P which uses both within-class alignment and between-class separateness (refer to line155 in the main article), Cosine loss uses the within-class alignment only which has faster NC and an advantage in Mixup training (refer to the empirical results in the next Response).
>
> $\textbf{Response to weakness 2}:$
>
> Thank you for your question. We did not pay enough attention to the empirical results w.r.t the loss P, since our major concern is the final state of the neural collapse and cosine loss is obviously simpler than P. We now provide a few experimental results w.r.t P, where the classifier is fixed. The table below shows the performance of P on cifar100 with different imbalance ratios.
>
> |$\tau$|0.005|0.01|0.02|
> |--------|--------|--------|--------|
> |P w/o mixup|  41.9$\pm$0.2|  43.4$\pm$0.3  |  43.5$\pm$0.2|
> |P w/ mixup|  36.5$\pm$0.6  |  40.1$\pm$0.4  |49.0$\pm$0.1|
> |CAL mixup|$\textbf{46.5}$$\pm$0.5| $\textbf{50.1}$$\pm$0.3| $\textbf{54.3}$$\pm$0.4|
>
> The performance of P is lower than CAL, and P starts to be incompatible with Mixup training when the ratio decreases (to 0.01). One possible explanation is that the Cosine loss do not have distraction terms from the centers of other classes, while P needs to handle the between-class separateness, which can become a complicated procedure when there are a myriad of centers presented.
>
>
>
> $\textbf{Response to weakness 4}:$
>
> We add the comparison to the method Rotated Balanced Representation (RBL) of [1] to our paper, and the literature will be discussed in the introduction or related work. It can be observed in the following table (the first four columns are cifar10 test accuracy and the last four are cifar100) that the learnable orthogonal layer is effective when the post-hoc logit adjustment (PLA) is applied (the ablation shows how powerful the post-hoc logit adjustment is). "-" indicates no results recorded (due to time limit).
>
> |$\tau$|0.005|0.01|0.02| 0.1 |0.005|0.01|0.02| 0.1 |
> |-----------|-------|-------|-------|-------|-------|-------|-------|-------|
> |RBL w/o PLA|73.6|78.5|84.3|90.7|-|-|-|-|
> |RBL w/ PLA|$\textbf{81.8}$$\pm$0.5|$\textbf{84.9}$$\pm$0.3|$\textbf{87.6}$$\pm$0.2|$\textbf{92.5}$$\pm$0.3|41.7$\pm$0.4|$\textbf{51.7}$$\pm$0.2|52.4$\pm$0.2|$\textbf{68.4}$$\pm$0.1|
> |CAL|80.0$\pm$0.5| 84.1$\pm$0.3| 85.9 $\pm$0.2| 92.0$\pm$0.3| $\textbf{46.5}$$\pm$0.5| 50.1$\pm$0.3| $\textbf{54.3}$$\pm$0.4| 65.9$\pm$0.3|
>
> Our method outperforms RBL in two settings for cifar100. We conjecture that in a setting of a small imbalanced ratio and a large number of classes, the hard-to-predict distribution may dominate the performance. The reason that our method CAL has lower accuracy compared to the method of [1] in other settings, is three-fold: 1. Imbalanced learning with MSE loss is less effective than CE loss in general; 2. Our experiment setting is chosen as close as possible to that where the theoretical analysis (proposition B.1 and C.1) is conducted, for example, to match the isotropy of the Gaussian, we normalize/batch normalize the feature to ensure it is unit norm and centered before the classification, and the weight are unit vectors through training. This setting possibly harms the learnability and flexibility of our model. 3. Under the class-aware MSE loss, we use the original classifier to be the surrogate classification rule of our general classification rule. However, the rule is designed especially for “hard-to-predict” unseen data and thus is not necessarily optimal for the classification of other unseen data. When the hard-to-predict unseen data takes a very small portion of the population, our design may lose its effectiveness. The success of RBL and PLA inspires us to find an optimal classifier for the loss P.

---

> > ### Author Response · Authors · 2024-08-10
> > **We appreciate your higher rating.**
> >
> > We appreciate your higher rating.

---

### Official Review · Reviewer_vNc6 · 2024-07-10

**Soundness:** 3
**Presentation:** 2
**Contribution:** 3
**Rating:** 6
**Confidence:** 3

**Summary:**

This paper addresses the issue of minority collapse in imbalanced learning, finding an optimal structure to represent a better classification rule. The authors induce a new definition called NCMC and design an MSE-type loss to alleviate the minority collapse phenomenon.

**Strengths:**

This paper is well-written and has a clear logic. It discusses an interesting phenomenon where features in the minor class contribute to mitigating the minority collapse, providing a novel perspective.

The authors also designed a practical loss function to induce NCMC and improve generalization.

**Weaknesses:**

The experimental results did not show significant improvement; the accuracies of these loss functions are quite similar. For example, in Table 4's CIFAR-100 experiments, the results might all be within the margin of error, suggesting that the newly proposed loss function may not be very effective.

**Questions:**

- Could the authors provide more visualizations of multiple centers?

- In the experiments, what is $\tau$ and $\theta$ refer to? How is the degree of imbalance discussed?

- Is the conclusion only applicable to MSE loss, or could it also apply to CE loss or other loss functions designed to handle imbalanced data?

- Is the NCMC phenomenon, like NC, model-agnostic? Would different backbones affect the conclusion?

---

> ### Author Rebuttal · Authors · 2024-08-07
>
> $\textbf{Response to weakness 1}$:
>
> The main purpose of this paper is to assign a novel classification rule to the imbalanced classification. The classification rule focuses on the hard-to-predict subpopulation, which is different from the popular margin theory.
>
>  In order to justify the usefulness of our theoretical result of the generalized classification rule, we compare the proposed class-aware MSE loss to several baselines. The most useful baseline is SETF, which has nearly the same settings as ours: SETF and CAL both consider fixed classifiers, use only the within-class alignment part of the MSE loss function, and apply the same training schedule. They are comprehensively compared on (four datasets $\times$ three imbalanced ratios $\times$ two backbone networks) (24 settings in total). CAL outperforms SETF in most settings, indicating that assigning more directions for minor classes is a useful strategy aiming more than alleviating minority collapse, and the generalized classification rule is effective. In particular, under either loss, all samples from a class collapse to only one vector, so CAL and SETF should have the same capability of reducing the minority collapse. The significant difference comes primarily from the number of directions: In the supervised training, SETF only has one available direction (the classifier vector) for the feature to align while CAL uses information from multiple directions (the centers of each class).
>
> We further examine the different choices of parameter $f$ and $\theta$; it is worth noting when $\theta = \frac{\pi}{2}$, i.e. the centers are all identical to their corresponding classifier vector, and the performance is no better than SETF. This observation shows reweighting alone does not improve the classification, thus our method is different from reweighting methods.
>
> There are comparisons to other classical methods, the comparable performance (or the marginal improvement) implies fixing classifier has the potential to be developed further and applied in practice.
>
> $\textbf{Response to question 1}$:
> The 3D illustration (Fig.1) of the multi-center frame is presented in the attached pdf. Imagine that all solid-line vectors are orthonormal vectors, among which the red and yellow ones are the $w_1$ and $w_{2}$ in the definition 3.2 which corresponds to class 1 and class 2. The solid green and solid blue are the augmented vectors associated with $w_1$ and $w_2$ resp. Then the green dashed-line vectors are the centers of class 1 and the blue ones are the centers of class 2. Since the classes has more than one center in general, we call them multiple centers.
>
> $\textbf{Response to question 2}$:
> We apologize that $\tau$ is not claimed in the main article before being used. We consider long-tail image data which is generated from imbalance sampling from the data including cifar10, cifar100, SVHN, and STL10. $\tau:= \frac{n_{min}}{n_{max}} \leq 1$ is the imbalance ratio that describes the exponential decay of class sample size $n_k$, more precisely, let $n_1>n_2>\ldots>n_K$, then $n_k = n_{k-1}\tau^{\frac{1}{K-1}}$; $\theta$ is defined in definition 3.2, which is the angle between $\mathbf{w}_{k}$ and its associate centers $\mathbf{w_j^{(k)}}$. Please see more discussions of the experiment settings in Appendix F and the 3D illustration in the attached pdf.
>
> $\textbf{Reponse to question 3}$:
> The logic here is, that the paper finds that a model needs more directions for minor classes (General Classification Rule, GCR), and then proposes loss P and Cosine Loss that lead to a surrogate of the GCR. In particular, we design the MSE-type loss to fit the purpose. It is highly possible to find an analogous conclusion under CE loss or other popular loss [1] if we can figure out the global optimal solution to the corresponding UFM under such losses. Our work demonstrates the effectiveness of the UFM analysis and the methodology can be useful to the model design: given the specific classification rule or other purposes, the joint structure of classifier and feature can be designed.
>
> $\textbf{Response to question 4:}$
>
> The Discovery of NC phenomenon [2] and the Classic analysis of UFM [3] tell us that NC is model-agonistic conditional on the high expressivity of the deep neural networks. What we know is NC will not happen if the network does not interpolate the data. We show NCMC occurs through UFM analysis, and it can be verified on loss P and the Cosine Loss. Fig.2 in the attached pdf shows the NC on different backbones. It is observed that ResNet, Densenet, and VGG have more severe collapse than LeNet, a relatively small model that hardly interpolates cifar10 in 200 epochs. However, formulating a complete theoretical justification of the occurrence of either NC or NCMC during SGD training is a hard yet ongoing problem in deep learning theory.
>
> [1] Zhou et al. Are All Losses Created Equal: A Neural Collapse Perspective. NeurIPS 2022.
>
> [2] Papyan et al.. Prevalence of neural collapse during the terminal phase of deep learning training. PNAS, volume 117, 2020.
>
> [3] Zhu et al.. A geometric analysis of neural collapse with unconstrained features. NeurIPS 2021.

---

> > ### Comment · Reviewer_vNc6 · 2024-08-13
> > **Response**
> >
> > Thanks for the detailed response. I will raise my score to 6.

---

> > > ### Author Response · Authors · 2024-08-13
> > >
> > > We are grateful for the reviewer's reconsideration.

---

### Official Review · Reviewer_HFbj · 2024-07-10

**Soundness:** 3
**Presentation:** 4
**Contribution:** 4
**Rating:** 7
**Confidence:** 4

**Summary:**

This paper studies the Neural Collapse (NC) phenomenon in imbalanced learning. Specifically, the authors find that the minor classes should align with more directions to achieve better classification results. Such finding yields the Generalized Classification Rule (GCR). The authors study NC under UFM. They find that the features of a class $k$ tend to collapse to the mean of centers of class $k$ which is termed Neural Collapse to Multiple Centers (NCMC), and RCR (the original classifier) approximates GCR at NCMC. Based on the above studies, the authors propose Cosine Loss and show from experiments that Cosine Loss can induce NCMC and has comparable performance to classical long-tail learning methods.

**Strengths:**

Overall, this paper is well-written and quite novel. Here is a detailed assessment:

1. **Originality**: The originality of the paper is commendable. This paper proposes a new classification rule named Generalized Classification Rule (GCR) for imbalanced learning, introduces Neural Collapse to Multiple Centers (NCMC) within UFM framework, and shows that the traditional RCR classification rule resembles GCR at NCMC. Based on such theoretical findings, this paper introduces a new type of loss termed Cosine Loss. Extensive studies show the effectiveness of Cosine Loss.
2. **Quality**: This paper is in good quality. The theoretical study is rigorous and quite convincing. Backed by the theoretical study, the effectiveness of proposed Cosine Loss has been verified by extensive experiments, providing strong evidence for their conclusions.
3. **Clarity**: The writing is clear and concise. From GCR to NCMC, then the resemblance of RCR to GCR at NCMC, finally the Cosine Loss and extensive experimental verification, the paper is well-organized and quite natural.
4. **Significance**: This paper proposes a new phenomenon called NCMC which deepens the understanding of Neural Collapse especially under imbalanced settings. Also, the proposed Cosine Loss is an effective long-tail learning method.

**Weaknesses:**

This paper is generally well-written without much weaknesses. Here are a few possible points.

1. The introduction of $\widetilde{\boldsymbol{w}}_j^{(k)}$ is a bit abrupt at line 121-124. I would suggest more explanations including in the corresponding appendix section. Also, the tilde symbol is missing in Eq.(8).
2. Some minor typos. Line 95 “denote the (mean?) of features of class $k$…”. Line 96 “$\boldsymbol{h}_k:==$” double =’s. Line 104 “$\{\boldsymbol{h})\}$” redundant ‘)’. End of line 138 “satisfies” to “satisfy”. Line 197 “approximates” to “approximate”.
3. Citation recommendations. I believe your work would benefit from referencing some additional literature to provide a more comprehensive context for your study. Specifically, i recommend citing the following articles:
  -  A Unified Generalization Analysis of Re-Weighting and Logit-Adjustment for Imbalanced Learning (NeurIPS 23)
  - Understanding imbalanced semantic segmentation through neural collapse (CVPR 23)
  - Deep long-tailed learning: A survey (TPAMI 23)
  - Harnessing Hierarchical Label Distribution Variations in Test Agnostic Long-tail Recognition (ICML 24)

**Questions:**

One question: why study under UFM? Is this generalizable?

**Limitations:**

See *Weaknesses*.

---

> ### Author Rebuttal · Authors · 2024-08-06
>
> Thank you for the recognition of our contribution.
>
> $\textbf{Response to the weakness}$:
>
> 1. thank you for your advice. We think it is a good idea to move proposition B.1 to the Main Result section and add more explanations of $\tilde{w}_j^{(k)}$. In particular, the set of $\tilde{w}_j^{(k)}$’s form an orthogonal frame (vectors are mutually orthogonal and have unit norm).
>
> 2. you are right about the typos, we will check over the formulas and words to guarantee the readability of this paper.
>
> 3. Thank you for the recommendations. The citations are recent literature on imbalanced learning and neural collapse. They are closely related to our work, and we will discuss their methods in the introduction and related work.
>
> $\textbf{Response to Question 1}$:
>
> We study UFM for the following two reasons:
>
> 1. Deep models are generally hard to analyze due to their intricacy and diversity. UFM offers an intuitive explanation of what representation has been learned and how the classification is carried out.
>
> 2. Because of the simplicity of UFM, it can be used to help design special representation structures and decision rules of interest, and the conclusions based on UFM are easy to verify.
>
> The underlying assumption of the effectiveness of UFM is the high expressivity of the deep neural networks. The conclusions such as the optimal structure of the classifier and the large margin analysis at the terminal phase of training based on UFM can be generalizable when the model is overparameterized and easily interpolates the data.
>
> Despite the power of UFM, it is no more than an expedient approach. Completely understanding the neural collapse and its connection to other deep learning phenomenons such as benign overfitting and grokking demands deeper theories.

---

> > ### Comment · Reviewer_HFbj · 2024-08-11
> >
> > Thank you for your response. I would keep my scores.

---

> > > ### Author Response · Authors · 2024-08-11
> > >
> > > Thank you for your feedback.

---

### Official Review · Reviewer_FHuh · 2024-07-10

**Soundness:** 3
**Presentation:** 2
**Contribution:** 3
**Rating:** 5
**Confidence:** 3

**Summary:**

This paper studies the Neural Collapse phenomenon under the imbalanced training data. The authors extend the optimal structure of neural collapse classification to a multiple center setting to enhance the model performance. Specifically, the authors propose to leverage the Generalized Classification Rule to make the minor classes align with more directions. Moreover, a practical MSE-type objective function has been proposed to train a "neural collapse to multiple centers" model. The proposed method achieves comparable performance with existing baselines.

**Strengths:**

The authors have conducted a thorough theoretical and experimental analysis on a list of SOTA neural collapse-related methods.

**Weaknesses:**

- The performance improvement of the proposed CAL in Table 4 is marginal in comparison to other baselines.

- The writing of the paper can be further improved.
  - The challenge and motivation of this work hasn't been well addressed in the introduction. Content in line 43-63 is more likely to appear in related work rather than introduction.
  - It is hard for readers to understand the meaning of a sentence when too many abbreviated words come together, e.g., at line 73-77.

**Questions:**

Please refer to the weakness section.

**Limitations:**

The authors have discussed the limitation of the proposed method, including the assumption made in the theoretical analysis, the computational cost and the not impressive performance on high-dimensional datasets.

---

> ### Author Rebuttal · Authors · 2024-08-06
>
> Thank you for your review.
>
> $\textbf{Response to weakness 1}$:
>
> As far as we see, the proposed CAL has marginal improvement on cifar10 compared to ARBloss, but clearly outperforms other classical methods in comparison; it also has non-negligible improvements on cifar100. The experiment shows that CAL is effective for the fixed classifiers under MSE loss, which matches our theoretical analysis from the perspective of Neural Collapse.
> There are quite a few SOTA results in imbalanced learning that outperforms our method, but our primary contributions are more than the performance improvement:
>
> (i) We formalize a novel principle we call “generalized classification rule” that focuses on the hard-to-predict subpopulation of the underlying data distribution, which is a different perspective compared to the popular margin theory and minority collapse. This rule claims that more directions are needed for minor classes in the imbalanced classification. The analysis is inspired by the recently discovered neural collapse phenomenon that for a highly expressive neural network, the classifier and penultimate layer feature has symmetric structures.
>
> (ii) We design the loss that leads the model to a state where the analysis in (i) is effective. In particular, since directly using GCR rule results in gradient vanishing and quick overfitting, we propose to study the surrogate objective P. We perform a theoretical analysis based on unconstrained features model to show that P leads to a neural collapse where the original classifier can be considered a representative of the multiple directions. In practice, we use Cosine Regression loss with a fixed classifier for data training. In contrast to P, Cosine regression loss has similar neural collapse types according to our theory (corollaries for fixed W), but is simpler and easier to train.
>
> (iv) The theoretical analysis is justified. The baseline SETF has nearly the same setting except that we use multiple directions for a class and normalize the features. Our method outperform SETF under most settings, which indicate that the information from multiple directions is useful for the imbalanced classification. This paper demonstrates the power of neural collapse analysis: through Unconstrained feature model we are able to design specific convergence pattern of the classifier and feature.
>
> In short, our work can provide several insights in imbalanced learning and general model design, and we hope the reviewer can re-evaluate our contributions.
>
> $\textbf{Response to weakness 2}$:
>
> a) Indeed, the paragraphs in line 43-63 refer to several closely related works to our paper, but they are also the inspiration of our work and serve as the intro to the presentation of our motivation and contributions in the face of the issue that NC does not offer enough information of the underlying data distribution. Since our narrative is not effective according to the reviewer, we will highlight our challenge and motivation more concisely in the revision and offer more discussions about these references in “Related Work” section.
>
> b) We are sorry that there are 4 abbreviated words between line 73 and line 77: UFM (unconstrained feature model), NCMC (Neural Collapse to Multiple Centers), RCR (Regular Classification Rule) and GCR (Generalized Classification Rule). We will reduce the frequency of abbreviate names per sentence in the revision. The whole paper will be carefully polished for high readability.

---

> > ### Comment · Reviewer_FHuh · 2024-08-11
> >
> > Thanks for the authors' response, which has addressed most of my concerns. I will raise my score to borderline accept. I hope the authors can further revise the paper as they promised in the rebuttal.

---

> > > ### Author Response · Authors · 2024-08-11
> > >
> > > Thank you for your re-evaluation of our work. We will revise the paper carefully according to the rebuttal.

---

### Author Rebuttal · Authors · 2024-08-07

Thanks to the reviewers for their patience and time.

According to the questions from the reviewers, we add a few experiments w.r.t loss P and the generalized classification rule. The attached contains three figures:

$\textbf{Figure 1}$: The 3D illustration of multi-center frame;

$\textbf{Figure 2}$: NCMC phenomenon in Different Backbones. We draw the mean and standard deviation of the neural collapse metric used in the paper for four different backbones that are trained on Cifar10-LT;

$\textbf{Figure 3}$: The NC Analysis of Loss P on different $\tau$ w/ or w/o regularization on the feature norm

Two of the four reviewers recognize our work as excellent, and the rest believe that the weakness of this paper is the marginal performance improvement to some baseline in our experiments. Although performance is a fairly important dimension of our work, we still hope that the other contributions including the theoretical novelty can be correctly evaluated. Let me restate here the contributions of this paper, which conform to the presentation in the main article.

(1) We formalize a novel principle we call the “generalized classification rule” that focuses on the hard-to-predict subpopulation of the underlying data distribution, which is a different perspective compared to the popular margin theory and minority collapse.

(2) We design the loss P, and perform a theoretical analysis based on unconstrained features model (UFM) to show that P and its variants (including the Cosine Regression Loss) lead to a neural collapse where the original classifier can be considered a representative of the multiple directions.

(3) The effectiveness of our loss function for fixed classifier can be justified by the empirical results.

---

### Decision · Program_Chairs · 2024-09-25

**Decision:**

Accept (poster)

**Comment:**

This paper studies the Neural Collapse (NC) phenomenon in imbalanced learning and finds that minor classes should align with more directions for improved classification results.

Initially, this paper received mixed scores, but the authors effectively addressed these concerns in their rebuttal. Consequently, I recommend accepting the paper. Nonetheless, the authors are required to follow the reviewers' suggestions and integrate their rebuttal content into the original paper.